# DNA barcoding and species delimitation of the Old World tooth-carps, family Aphaniidae Hoedeman, 1949 (Teleostei: Cyprinodontiformes)

**Hamid Reza Esmaeili**[1]*, **Azad Teimori**[2], **Fatah Zarei**[1], **Golnaz Sayyadzadeh**[1]

**1** Ichthyology and Molecular Systematics Research Laboratory, Zoology Section, Department of Biology, College of Sciences, Shiraz University, Shiraz, Iran, **2** Department of Biology, Faculty of Sciences, Shahid Bahonar University of Kerman, Kerman, Iran

* hresmaeili@shirazu.ac.ir

**Data Availability Statement:** All DNA sequences files are available from the Genbank database. Accession numbers are provided in Table 1. Specimens studied are available and can be

## Abstract

The fishes, which have currently named *Aphanius* Nardo, 1827 are the relict of the ancient ichthyofauna of the Tethys Sea. For a long time since 1827, the genus name has been subjected to revision by several researchers using mainly morphological features. Until recently, no comprehensive single- or multi-locus DNA barcoding study has been conducted on whole members of the family Aphaniidae. In the present study, by applying four conceptually different molecular species delimitation methods, including one distance-based method, one network-based and two topology-based methods, we examined a single-locus DNA barcode library (COI) diversity for the 268 sequences within the family Aphaniidae from the Old World (57 sequences are new in the present study and 211 sequences were retrieved from NCBI database). The molecular analyses revealed a clearer picture of intra-family relationships and allowed us to clarify the generic names, and also describe a new genus for the family Aphaniidae. Results supported distinction of three major clades related to three genera within this family: i) the first clade includes the *A. mento* group which are placed in a new genus, *Paraphanius* gen. nov., found in the Orontes (= Asi) and Tigris-Euphrates River drainage, the Levant in coastal waters and the Dead Sea basin, western Jordan, and in southern Turkey in the Mediterranean basins as well as in central Turkey. This clade positioned at the base of the phylogenetic tree, (ii) the second clade contains the *A. dispar*-like brackish water tooth-carps which are transferred to the genus *Aphaniops* Hoedeman, 1951 (type species, *Lebias dispar*), distributed in the coastal waters around the Red Sea and the Persian Gulf basins; and (iii) the third clade, the genus *Aphanius* Nardo, 1827 (type species *Aphanius nanus* = *A. fasciatus*) contains all the inland and inland-related tooth-carps, which are mainly distributed in the inland waters in Turkey and Iran and also in the inland-related drainages around the Mediterranean basin.

accessed (deposited in the museums mentioned in the paper).

**Funding:** The research work was funded by Shiraz University provided to the first Author.

**Competing interests:** The authors have declared that no competing interests exist.

## Introduction

Morphological characters have long been used to describe and identify fish species. Nevertheless, it is known that traditional methods based on morphological properties often do not allow the discovery of cryptic species, frequently leading to taxon misidentification, instability in the number of species and an insufficient indication of the loss of genetic diversity. Since all fields of biology depend on the convenient species identification [1, 2], the species delimitation needs to be efficient and reliable.

DNA barcoding based on the partial nucleotide sequences of the cytochrome c oxidase I (COI) gene has been increasingly used as one of the principal approaches in the bio-identification system of animals [3]. Almost all species can be distinguished by their COI sequences only if the average intraspecific and interspecific genetic distances are significantly different (barcoding gap) [4, 5].

The most important benefits of DNA barcoding are its accuracy to distinguish closely related species, identifying individuals at different developmental stages, discovering cryptic species and to identify the possible causes of synonymies. Therefore, over recent decades, this technology has had a principal contribution to the reliable identification of species [6, 7].

The Old World Cyprinodontiformes (now in the genus *Aphanius* Nardo, 1827), including the extant and fossil species, are widely distributed along the late-period Tethys Sea coastlines. It is known that differences in the Mediterranean Sea level created by glacial and interglacial conditions have largely affected their present-day distribution. Since the tooth-carp fishes are not highly diverse in external morphology, therefore, primary morphological attempts were not quite successful for species identification. For this reason, the systematics of these fishes has been the subject of many studies since the first description of the *Aphanius* species in 1821 by Valenciennes.

### What is *Aphanius* Nardo, 1827?

The fishes that have currently named *Aphanius* Nardo, 1827 (ref. 1827a, pp. 34, 39–40; also published in 1827b, col. 487), are the relict of the ancient ichthyofauna of the Tethys Sea. These fishes first appeared in the middle to late Aquitanian. The oldest records, described based on the otolith morphology are from Germany (the Mainz Basin) and SW France (the Aquitaine Basin), 'genus Cyprinodontidarum' [= *Aphanius*] *angulosus* Steurbaut, 1978) (cf. Reichenbacher 2000). *Aphanius princeps*, found in the deposits of the Burdigalian age in Catalonia, NE Spain, is the taxon with the oldest fossil skeleton [8].

The genus *Aphanius* was the only native representative of the family Aphaniidae in the Old World formerly considered to be in the family Cyprinodontidae [9, 10–12]. It was Sethi [13] who for the first time, advocated placing *Aphanius* in a distinct family, Aphaniidae, however, this suggestion did not find general acceptance. Typically, the name *Aphanius* refers to a genus of fresh and brackish water tooth-carps with a wide distribution, basically peri-Mediterranean, extending from the Mediterranean basin including Portugal and Morocco to the Persian Gulf and its eastern regions, including Pakistan and India [9, 14–18]. These fishes are generally small (up to 70 mm long) thriving in coastal and brackish water environments, the inland and land-locked water systems such as spring-stream systems, hot sulphuric springs, and rarely wetlands [16, 19–23].

**Systematics and nomenclature history.** Since 1827, the genus name has been subjected to revision by several researchers (see Kottelat and Wheeler [24]) and it has been widely used since at least 1926. Originally, the genus was established with two nominal species *Aphanius nanus* Nardo, 1827 and *Aphanius fasciatus* Nardo, 1827 from the Mediterranean waters. Later *A. nanus* was selected as a junior synonym of *Lebias fasciata* Valenciennes in Cuvier &

Valenciennes, 1827 (Jordan [25], p. 121). The type species, therefore, becomes "*fasciatus*", now *A. fasciatus* by the subsequent designation of Jordan [25].

*Aphanius* has been named by some authors as a junior synonym of *Lebias* Goldfuss, 1820 [26]. It should be noted that for more than 150 years *Lebias* had been considered as a junior synonym of *Cyprinodon* Lacépède, 1803. However, with a single exception in 1895, it remained unused since 1846. In addition, Hoedeman [27] has examined four specimens of the genus *Tellia* from the collection in the Paris Museum and proposed that *Tellia* should not be separated from *Aphanius*, while *Aphanius dispar* is quite well separable from *Aphanius*. Therefore, a new generic name *Aphaniops* ("*Aphaniops* = *Aphanius*-opsis; resembling *Aphanius*", anticipated by Hoedeman [27]) was proposed for *A. dispar*. The new genus *Aphaniops* Hoedeman, 1951 was based on the following characters; having no dermal sheath around the anterior anal-fin rays, presence of 8–9 dorsal-fin rays in *Aphaniops* versus 10–14 dorsal-fin rays in *Aphanius*, 7–8 pelvic-fin rays in *Aphaniops* in contrast to 5–7 pelvic-fin rays in *Aphanius*. Accordingly, he provided the following key for the discrimination of *Aphanius* from *Aphaniops*:

1. "Dorsal fin with 8–9 rays, ventral fins well developed with 7–8 rays, no dermal sheath around the anterior anal rays, and length up to 80 mm. Genus *Aphaniops*.

2. Dorsal fin with 10–14 rays; ventral fins well developed, rudimentary or entirely wanting, with 5–7 rays if present, the presence of the dermal sheath around first few anal rays, naked or covered with scales, and length up to 65 mm. Genus *Aphanius*".

Also, according to Parenti [28], *Aphanius mento* possesses several diagnostic features not found in other members of the genus *Aphanius*, including a cartilaginous interhyal (ossified in other *Aphanius* species), an embedded urohyal (not embedded in other *Aphanius*), an upturned lower jaw (not upturned in other *Aphanius* species), and a distinctive neuromast pattern on the dorsal surface of the head (less prominently developed in other *Aphanius* species). As a result, she interpreted *A. mento* as the derived member of the *Aphanius* clade that should be designated as "*Aphanius*". This taxonomic separation corresponds to the observation of the sulcus morphology of sagittal otolith in the study of Reichenbacher et al. [29]. In the following, the list of synonyms is provided for the genus *Aphanius*:

*Aphanius Nardo, 1827*. Synonym of *Lebias* Goldfuss 1820, according to Lazara [30, 26]. Valid as *Aphanius* Nardo 1827, according to Tortonese [31], Parenti [28], Wildekamp et al. [32], Coad [33], Kruppand Schneider [34], Wildekamp [16], Poll & Gosse [35], Seegers [36], Wildekamp et al. [23], Doadrio et al. [37], Huber [38], Blanco et al. [39], Hrbek et al. [40], Kottelat et al. [41], Kottelat & Freyhof [42], Hertwig [43], Coad [44], Teimori et al. [45], Esmaeili et al. [46], Teimori et al. [47], Esmaeili et al. [19], Gholami et al. [48], Keivany & Esmaeili [49], Esmaeili et al. [50], Teimori et al. [51], Pfleiderer et al. [52], Huber [53], Jouladeh-Roudbar et al. [54], Esmaeili et al. [55], Freyhof et al. [9, 10], Çiçek et al. [56], Esmaeili et al. [57], Golani & Fricke [58], Teimori et al. [15, 59], Yoğurtçuoğlu & Freyhof [60], Teimori & Motamedi [61], and Motamedi et al. [62].

*Anatolichthys Kosswig & Sözer, 1945*. Valid subgenus, according to Lazara [30].
*Anatolichthys splendens* Kosswig & Sözer, 1945.
Synonym of *Kosswigichthys* Sözer, 1942, according to Parenti [28]. Synonym of *Aphanius* Nardo, 1827, according to Wildekamp [16], Wildekamp et al. [23], a synonym of *Lebias* Goldfuss, 1820.

*Aphaniops Hoedeman, 1951*. Valid subgenus, according to Lazara [30].
*Lebias dispar* Rüppell, 1829.

Synonym of *Lebias* Goldfuss 1820. Synonym of *Aphanius* Nardo, 1827, according to Tortonese [31], Parenti [28], Wildekamp et al. [32], Krupp & Schneider [34], Wildekamp [16], and Wildekamp et al. [23].

*Kosswigichthys Sözer*, *1942*. Valid subgenus, according to Lazara [30].

*Kosswigichthys asquamatus* Sözer,1942.

Synonym of *Lebias* Goldfuss 1820. Synonym of *Aphanius* Nardo, 1827, according to Wildekamp [16] and Wildekamp et al. [23].

*Lebias Goldfuss*, *1820. Lebias banded* Valenciennes, 1821. Valid as *Lebias* Goldfuss, 1820, according to Lazara [26], Costa [63] and Lazara [30]. The invalid generic name, according to Huber [38].

Synonym of *Aphanius* Nardo, 1827, according to Parenti [28].

*Micromugil Gulia*, *1861*. Valid subgenus, according to Lazara [30].

Synonym of *Aphanius* Nardo, 1827, according to Tortonese [31], Parenti [28], Wildekamp et al. [32], Wildekamp [16] and Wildekamp et al. [23]. Synonym of *Lebias* Goldfuss, 1820.

*Tellia Gervais*, *1853*. Valid subgenus, according to Huber [38].

Synonym of *Aphanius* Nardo, 1827, according to Parenti [28], Wildekamp et al. [32], Wildekamp [16] and Wildekamp et al. [23]. Synonym of *Lebias* Goldfuss, 1820.

*Turkichthys Ermin*, *1946*. Valid subgenus, according to Lazara [30].

*Turkichthys transgrediens* Ermin, 1946. Synonym of *Aphanius* Nardo, 1827, according to Wildekamp [16] and Wildekamp et al. [23]. Synonym of *Lebias* Goldfuss, 1820.

This background leads to the question: is *Aphanius* a monophyletic group? Parenti [28] has implied the paraphyly of *Aphanius* (the genus is probably not monophyletic). In addition, Nelson et al. [64] noted that the genus *Aphanius* probably is not a monophyletic group.

To date, 44 species have been described under the genus *Aphanius* [65]. The Near East, especially Iran and Turkey host the largest number of species [11, 16, 40, 44, 66]. As mentioned above, numerous authors have so far studied different aspects of these fishes, including the phenotypic variation, embryology, allopatric distribution, species diversity, historical zoogeography, evolutionary history and phylogenetic relationships [11, 23, 28, 67–79]. Particularly, significant progress has been made during the past decade to reconstruct the phylogenetic relationships of *Aphanius* species in the Near East [10, 15, 19–22, 40, 44, 46–48, 50, 51, 80–82]. Nevertheless, no comprehensive single- or multi-locus DNA barcoding study has been conducted for species delimitation in the family Aphaniidae.

In this study, by applying four conceptually different molecular species delimitation methods including one distance-based method, one network-based and two topology-based methods, we examined a single-locus DNA barcode library (COI) diversity for 268 sequences within the family Aphaniidae from the Old World (57 sequences are new in this study and 211 sequences retrieved from NCBI database). The aims of our study were (i) to use DNA barcoding as a tool for species delimitation within the family Aphaniidae, ii) to give a clearer picture of intra-family relationships and (iii) to clarify the generic names of the species group within the family. The DNA barcode records generated in this study are discussed by considering some morphological features of these fishes.

## Materials and methods

### Ethics statement

All fish species were caught in the inland water bodies (not national parks, other protected areas, or private areas, etc.), so no specific permissions were required for these locations/activities. Ethical approval was given by the Bioethics Committee of Biology Department, Shiraz University. Specimen collection and maintenance were performed in strict accordance

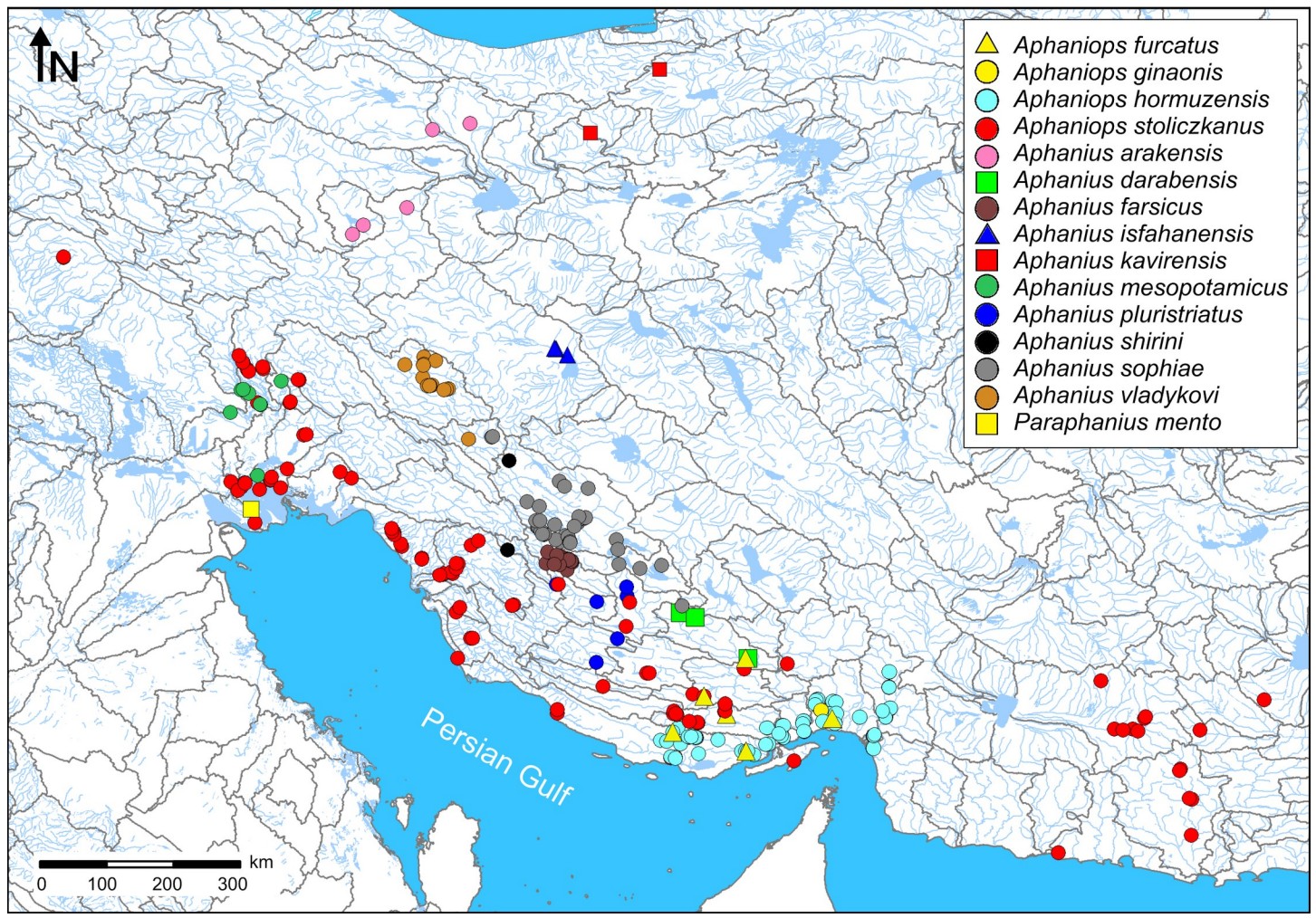

**Fig 1. Overview of the distribution areas and the sampling sites of the tooth-carps from Iran.** The map was originally designed in DIVA-GIS ver. 7.5 (https://www.diva-gis.org/).

with the recommendations of the Bioethics Committee of Biology Department, Shiraz University.

## Sampling and external morphological identification

A total of 57 specimens belonging to 14 species were sampled from various localities in Iran by using a hand net (Fig 1, Table 1). After anesthesia with 1% clove oil solution, these specimens were kept in 96% ethanol for molecular studies. The fish individuals were identified in the laboratory by two authors (H.R. Esmaeili and A. Teimori) based on the external morphology and including coloration, their geographic distribution and also by considering their original species descriptions and recently published articles [19, 33, 40, 44–48, 51, 83]. Meristic counts were done externally (not using cleared and stained specimens).

## DNA extraction, PCR and sequencing

In a sterile condition, small fragments of the dorsal muscle from alcohol-preserved specimens were sampled and placed in 96-well Eppendorf PCR plates. Genomic DNA of 57 specimens

**Table 1. New COI sequences for the tooth-carp species sampled from different drainage basins in Iran.**

| Species | Current proposed status | Basin/locality/coordinates | ZM-CBSU numbers | Accession No. |
|---|---|---|---|---|
| *A. vladykovi* | *Aphanius vladykovi* | Karoun River/Gandoman wetland/31˚ 50'05.0″N 51˚04'36.4″E | Ex85C9, Ex85C8, Ex85C10, Ex59G5 | MT102153, MT102154, MT102155, MT102156 |
| *Aphanius* sp. | *Aphanius* sp. | Sirjan/Nahr-e-Masih spring/30˚01'56.6″N 54˚19'54.5″E | ZG314, ZG313, ZG312, ZG311 | MT102157, MT102158, MT102159, MT102160 |
| *A. sophiae* | *Aphanius sophiae* | Kor River/ Marvdasht/29˚52'19.7″N 52˚29'48.4″E | ZG129, ZG128, ZG127 | MT102161, MT102162, MT102163 |
| | | Karoun River/Bibiseydan spring/31˚11'2.3″N 51˚26'59.2″E | ZG327, ZG326, ZG325 | MT102164, MT102165, MT102166 |
| | | Helleh/ Arjan wetland/29˚39'20.99″N 51˚59'14.39″E | ZG112, ZG109, ZG108 | MT102167, MT102168, MT102169 |
| *A. shirini* | *Aphanius shirini* | Kor River/Khosroshirin River/30˚53'29.5″N 52˚00'36.8″E | ZG272, ZG270, ZG267 | MT102170, MT102171, MT102172 |
| | | Helleh/ Arjan wetland/29˚39'20.99″N 51˚59'14.39″E | ZG113 | MT102173 |
| *A. pluristriatus* | *Aphanius pluristriatus* | Mond River/Jahrom/28˚26'N 53˚30'E | Ex90B8, Ex90B6, Ex90B11 | MT102174, MT102175, MT102176 |
| | | Mond River/ Khonj city/28˚06'32″N 53˚12'59″E | ZG120, ZG115, ZG114 | MT102204, MT102205, MT102206 |
| *A. mesopotamicus* | *Aphanius mesopotamicus* | Karoun River/ Jarahi River/30˚41'01.4″N 48˚32'22.7″E | ZG320, ZG319, ZG317, ZG315 | MT102177, MT102178, MT102179, MT102180 |
| *A. kavirensis* | *Aphanius kavirensis* | Kavir/ Cheshmeh-e-Ali spring/36˚16'45.45.6″N 54˚05'01.6″E | ZG141, ZG130, ZG1291, ZG1281 | MT102181, MT102182, MT102183, MT102184 |
| *A. isfahanensis* | *Aphanius isfahanensis* | Esfahan/Varzaneh, Zayandehud River/32˚25'26.28″N 52˚39'14.3″E | ZG342, ZG341, Ex53H4, Ex53H3 | MT102185, MT102186, MT102187, MT102188 |
| *A. ginaonis* | *Aphaniops ginaonis* | Hormuz/Genow spring/27˚26'51.0″N 56˚18'10.0″E | divmxext3 | MT102189 |
| *A. furcatus* | *Aphaniops furcatus* | Hormuz/Kol River/27˚38'19.0"N 54˚41'33.2″E | Ex88F2 | MT102190 |
| | | Hormuz/Khurgu spring/27˚31'21.3"N 56˚28'12.7"E | Ex88F1 | MT102191 |
| | | Hormuz/Faryab spring/27˚26'01.0"N 54˚16'43.0"E | Ex88E11 | MT102192 |
| *A. farsicus* | *Aphanius farsicus* | Maharlu Lake basin/Barmeshoor spring/29˚27'9.51″N 52˚42'0.05″E | ZG249, ZG140, ZG139, SExS02F7, SExS02F6, SExS02F5 | MT102193, MT102194, MT102195, MT102196, MT102197, MT102198 |
| *A. stoliczkanus* | *Aphaniops stoliczkanus* | Persian Gulf/ShourAb at Emamzadeh Shahid/28 32 35.3N, 52 21 43.8E | A621F | MT102199 |
| *A. darabensis* | *Aphanius darabensis* | Hormuz/Korsia-Banaki spring/28˚46'25″N 54˚23'48″E | ZG344, ZG137, ZG134, A622F | MT102200, MT102201, MT102202, MT102203 |
| *A. arakensis* | *Aphanius arakensis* | Namak Lake/Arak spring/34˚00'35.1″N 49˚50'50.8″E | ZG351, ZG349, ZG348 | MT102207, MT102208, MT102209 |

belonging to morphologically identified species based on external characters was purified by extraction from tissues in the presence of high concentrations of Guanidinium Thiocyanate and passage of the extracts through a fiberglass membrane (AcroPrep 1 μM glass fiber; PALL 5051) as described by Vargas et al. [84]. The standard DNA barcode region for vertebrates (COI) was amplified using the primer pair named FishF1 (5' TCAACCAACCACAAAGACAT TGGCAC3' ) and FishR1 (5' TAGACTTCTGGGTGGCCAAAGAATCA3' ) [4]. The amplification process was performed using Master Mix in a total volume of 25 μl containing 12.5 μl of a Ready 2X PCR Master Mix (Genetbio, Cat. no. G-2000), 0.5 μl of each primer (10 pmol/μl), 5 μl of the DNA template and 6.5 μl dd water. The amplification was performed on a Bioer XP Thermal Cycler (Bioer Technology Co. Ltd., Hangzhou, China), programmed as following: an initial denaturation at 94˚C for 3 min, 35 cycles with denaturation at 94˚C for 50 s, annealing

at 52˚C for 1min, and a final extension phase at 72˚C for 5 min. After purification with the ExoASP-IT® (usb) kit, the PCR products were Sanger sequenced with BigDye® Terminator v3.1 Cycle Sequencing Kit (Applied Biosystems, Foster City, CA) on an ABI PRISM 3730xl DNA Analyzer (Applied Biosystems, Foster City, CA) by the Macrogen Company of South Korea.

### Phylogenetic reconstruction

For obtaining a reliable outcome, a total of 211 COI sequences [10, 67, 85–88] from 28 *Aphanius* species were retrieved from GenBank and included in the phylogenetic analyses (Table 2). The goby killifish, *Aphyosemion franzwerneri* (Accession number: EF417044) was used as an outgroup. The BioEdit ver. 7.0.4 [89] was used to read the DNA chromatograms. The mtDNA sequences were first aligned with the ClustalW procedure implemented in MEGA ver. 7[90], and then were aligned manually. New sequences are deposited in NCBI Genbank (www.ncbi.nlm.nih.gov) under accession numbers MT102153-MT102209. We examined the substitution saturation with DAMBE ver. 7.2.7 [91]. The nucleotide substitution model best fitting the COI barcode library for the genus *Aphanius* was obtained using Akaike and Bayesian information criteria (AIC & BIC), and a decision theory method (DT) in JModelTest ver. 2.1.3 [92]. All criteria suggested GTR+I+G as the best-fit model for the sequence set. For phylogenetic reconstruction, the Bayesian method was run based on four simultaneous runs of four Markov chains for 100,000,000 generations and a burn-in of 15% of the initial trees in MrBayes ver. 3.2.6 [93]. An ultrametric gene tree was generated using the uncorrelated lognormal relaxed clock model [94] and the birth-death model in BEAST ver.1.8.2 [95]. BEAUTi ver. 1.8.2 [95] was used to implement the run settings: 100 million chain length, sampling each 3000th tree, the uncorrelated lognormal relaxed clock, birth-death model and GTR+I+G substitution model with four gamma categories. Tracer ver. 1.6 [96] was used to test the appropriateness of parameters. The maximum age for *Aphanius* was set at 34 Ma based on the oldest known fossil of the Old World killifishes, *Prolebias stenoura* Sauvage, 1874 [97]. Results were visualized in Tree Annotator ver. 1.8.2[98] with a 10% burn-in rate, 95% highest posterior density of divergence times and under the maximum clade credibility option for the consensus tree. In addition, we generated a Maximum Likelihood (ML) tree with 3,000 bootstrap replicates using RAxML ver. 7.2.5 [99] under the GTR+I+G nucleotide substitution model, with fast bootstrap.

### Molecular species delimitation

Four molecular species delimitation methods were applied to reduce difficulties when using only single parameter estimates and to compare results of conceptually different approaches, including a distance-based method, a network-based, and two topology-based approaches. The Automatic Barcode Gap Discovery (ABGD) method [100] uses the distribution of pairwise differences to detect a barcode gap dividing hypothetical species in the sequence set by assuring that intraspecific and interspecific distances do not overlap. We tested a set of COI sequence of the studied tooth-carps on the ABGD web server (https://bioinfo.mnhn.fr/abi/public/abgd/) with a combination of ABGD settings within the parameter range of Pmin = 0.001, Pmax = 0.1 and gap width = 0.1–1.2, all for a total of 20 steps and applying a Kimura-2-parameter (K2P) [101] corrected genetic distance matrix calculated using MEGA ver. 7.0 [90]. The reversed Statistical Parsimony (SP) method [102] is a network-based system that delineates hypothetical species using the network topology. The TCS ver. 1.21 [103] software was used to calculate a statistical parsimony network. The haplotypes are clustered into two separate networks or species if the number of mutations between two neighboring haplotypes, being more than the connection probability. We used a 95% connection probability

**Table 2. Details of 211 COI sequences belonging to 28 *Aphanius* species acquired from the GenBank database.**

| Species | N[a] | GenBank accession numbers | | | | | | |
|---|---|---|---|---|---|---|---|---|
| *A. alexandri* | 3 | KJ552602 | KJ552715 | KJ552647 | | | | |
| *A. almiriensis* | 8 | KJ552360 | KJ552520 | KJ552640 | KJ552735 | MH410024 | MH410025 | MH410026 |
| | | MH410027 | | | | | | |
| *A. anatoliae* | 10 | AY356565 | KJ552467 | KJ552604 | KJ552353 | KJ552362 | KJ552421 | KJ552426 |
| | | KJ552691 | KJ552704 | KJ552628 | | | | |
| *A. apodus* | 3 | KJ552553 | KJ552606 | KJ552719 | | | | |
| *A. baeticus* | 4 | KJ552741 | KJ552418 | KJ552475 | KJ552714 | | | |
| *A. dispar* | 32 | MF918578 | MG013854 | MG013855 | MG013857 | MF918579 | MF918580 | MG013846 |
| | | MG013847 | MG013848 | MG013849 | MG013850 | MG013852 | MG013853 | MG013859 |
| | | MG013860 | MG013861 | MG013862 | MG013863 | MF918576 | MF918577 | MG013864 |
| | | MG013865 | MG013866 | MG013867 | MF918581 | MF918582 | MF918583 | MG013874 |
| | | MG013873 | MG013875 | MF918588 | MF918589 | | | |
| *A. fasciatus* | 32 | KJ552453 | KJ552516 | MH410031 | MH410032 | KJ552372 | KJ552597 | KJ552687 |
| | | KJ552699 | MH492715 | KJ552449 | KJ552618 | KJ552621 | KJ552624 | KJ552744 |
| | | MH492716 | MH492717 | MH492718 | MH492719 | MH492720 | KJ552363 | KJ552667 |
| | | KJ552751 | KJ552757 | KJ552428 | KJ552559 | KJ552584 | KJ552610 | KJ552726 |
| | | KJ552524 | MH410028 | MH410029 | MH410030 | | | |
| *A. fontinalis* | 4 | KJ552742 | KJ552700 | KJ552560 | KJ552722 | | | |
| *A. ginaonis* | 3 | MF918590 | MF918591 | MF918592 | | | | |
| *A. iberus* | 7 | KJ552361 | KJ552729 | KJ552419 | KJ552617 | KJ552466 | KJ552653 | KJ552705 |
| *A. iconii* | 2 | KJ552481 | KJ552688 | | | | | |
| *A. kruppi* | 5 | MF918593 | MF918594 | MF918595 | MF918596 | MF918598 | | |
| *A. maeandricus* | 5 | KJ552515 | KJ552532 | KJ552605 | KJ552543 | KJ552645 | | |
| *A. mento* | 9 | KJ552373 | KJ552492 | KJ552576 | KJ552442 | KJ552750 | KJ552446 | KJ552673 |
| | | KJ552713 | KJ552409 | | | | | |
| *A. mentoides* | 5 | KJ552397 | KJ552456 | KJ552482 | KJ552668 | KJ552702 | | |
| *A. meridionalis* | 7 | KJ552432 | KJ552461 | KJ552533 | KJ552544 | KJ552660 | KJ552706 | KJ552743 |
| *A. orontis* | 2 | KJ552423 | KJ552683 | | | | | |
| *A. richardsoni* | 8 | KJ552395 | KJ552571 | KJ552638 | KJ552643 | KJ552736 | MF918599 | MF918600 |
| | | MF918601 | | | | | | |
| *A. saldae* | 2 | KJ552398 | KJ552630 | | | | | |
| *A. saourensis* | 5 | KJ552386 | KJ552497 | KJ552623 | KJ552670 | KJ552701 | | |
| *A. similis* | 4 | KJ552414 | KJ552500 | KJ552367 | KJ552393 | | | |
| *A. sirhani* | 3 | MF918602 | MF918603 | KJ552402 | | | | |
| *A. stiassnyae* | 5 | MG013868 | MG013869 | MG013870 | MG013871 | MG013872 | | |
| *A. stoliczkanus* | 23 | MF918615 | MF918616 | MF918611 | MF918612 | MF918613 | MF918614 | MF918617 |
| | | MF918618 | MF918619 | MF918620 | MF918621 | MF918622 | MF918605 | MF918606 |
| | | MF918607 | MF918608 | MF918609 | MF918624 | KU499803 | KU499804 | KJ552389 |
| | | KJ552732 | MF918622 | | | | | |
| *A. sureyanus* | 2 | KJ552468 | KJ552758 | | | | | |
| *A. transgrediens* | 8 | KJ552368 | KJ552404 | KJ552410 | KJ552452 | KJ552528 | KJ552574 | KJ552679 |
| | | KJ552710 | | | | | | |
| *A. villwocki* | 2 | KJ552538 | KJ552582 | | | | | |
| *Aphanius* sp. (Syria) | 8 | KJ552593 | KJ552483 | KJ552542 | KJ552692 | KJ552471 | KJ552536 | KJ552635 |
| | | KJ552711 | | | | | | |

[a]Number of sequences

threshold to delineate hypothetical species. The Bayesian Poisson Tree Process (bPTP) approach [104] uses a tree topology to delineate species. It assumes that each mutation event has a non-null probability of forming a new species and, thus, the number of interspecific mutations is significantly higher than the number of intraspecific mutations. The bPTP server (https://species.h-its.org/ptp/) was used with a Bayesian topology produced in MrBayes ver. 3.2.6 as input tree. The analysis was run under default settings. The Bayesian General Mixed Yule-Coalescent (bGMYC) method [105] is conceptually similar to bPTP and uses a tree topology to infer species hypotheses, but unlike bPTP, it applies an ultrametric tree as an input file. This method implies that each node of a gene tree corresponds to one of these two possible events: (i) divergence between species after a strict Yule process [106]; or (ii) neutral coalescent events of lineages forming a species [107]. Since the rate of coalescent events is higher than speciation, it is possible to define a limit on a phylogenetic tree between interspecific and intraspecific divergence, delimiting clusters of specimens, which are genetically isolated, and independently evolving lineages or species. This analysis was performed using the bGMYC package [105] for R ver. 3.6.1 [108]on an ultrametric tree produced in BEAST ver. 1.8.2 as an input file.

## Results

### Molecular species delimitation

The final COI alignment included 268 sequences and one out-group sequence (i.e., *Aphyosemion franzwerneri*) trimmed to 609 bps. The nucleotide substitution pattern showed that the sequences have not reached substitution saturation and are therefore well applicable for phylogenetic analyses. The data were analyzed using the Bayesian Inference and Maximum Likelihood (ML) methods (Fig 2). All the nominal species were recovered as monophyletic except *A. dispar* (Rüppell, 1829), *A. stiassnyae* Getahun & Lazara, 2001 (from Ethiopia, Lake Afrera), *A. pluristriatus* (Jenkins, 1910), and *A. fontinalis* Aksiray, 1948 (from Turkey). The phylogenetic position of *A. richardsoni* (Boulenger, 1907), *A. stiassnyae*, *A. kruppi*, *A. ginaonis* and *A. stoliczkanus* made *A. dispar* a paraphyletic species. In addition, the phylogenetic position of *A. sureyanus* (Neu, 1937) made *A. fontinalis* paraphyletic. The clade formed by *A. pluristriatus A. kavirensis*, *A. sophiae*, *Aphanius* sp. and *A. mesopotamicus* is plagued with very low levels of interspecific sequence divergence (S1 Table), and low values of posterior probabilities and bootstraps (Fig 2).

The four conceptually different molecular species delimitation methods revealed 35–48 species (Fig 2). The ABGD method delineated 40 species (Fig 2). Out of 40 nominal species, 26 (65%) were delimited through the ABGD tool. It indicated potential species-level diversity in five nominal species: *A. dispar*, *A. kruppi*, *A. ginaonis*, *A. stoliczkanus*, and *A. anatoliae*. ABGD presented the following groups of nominal species/sequences with low genetic diversity): (i) *A. stiassnyae* and *A. dispar* specimens from Lake-Assal (Djibouti), Afrera spring (North Ethiopia), Lake Abaeded (Ethiopia), Gali Colluli River (Eritrea), Zariga River (Eritrea), and Shukaray River (Eritrea; MF918580 and MF918579); (ii) *A. arakensis*, *A. pluristriatus*, *A. kavirensis*, *A. sophiae*, *Aphanius* sp. and *A. mesopotamicus* (all from Iran); and (iii) *A. fontinalis* and *A. sureyanus*.

The SP method delineated 35 species (Fig 2). Out of 40 morphologically identified species, 26 (65%) were delimited nicely through the SP tool. The SP analysis presented potential species-level diversity in two nominal species including *A. dispar* and *A. anatoliae*. This method also revealed low genetic diversity in the COI barcode region of the following species:(i) *A. stiassnyae* and *A. dispar* specimens from Lac-Assal (Djibouti), Afrera spring (North Ethiopia), Lake Abaeded (Ethiopia), Gali Colluli river (Eritrea), Zariga river (Eritrea), and Shukaray river

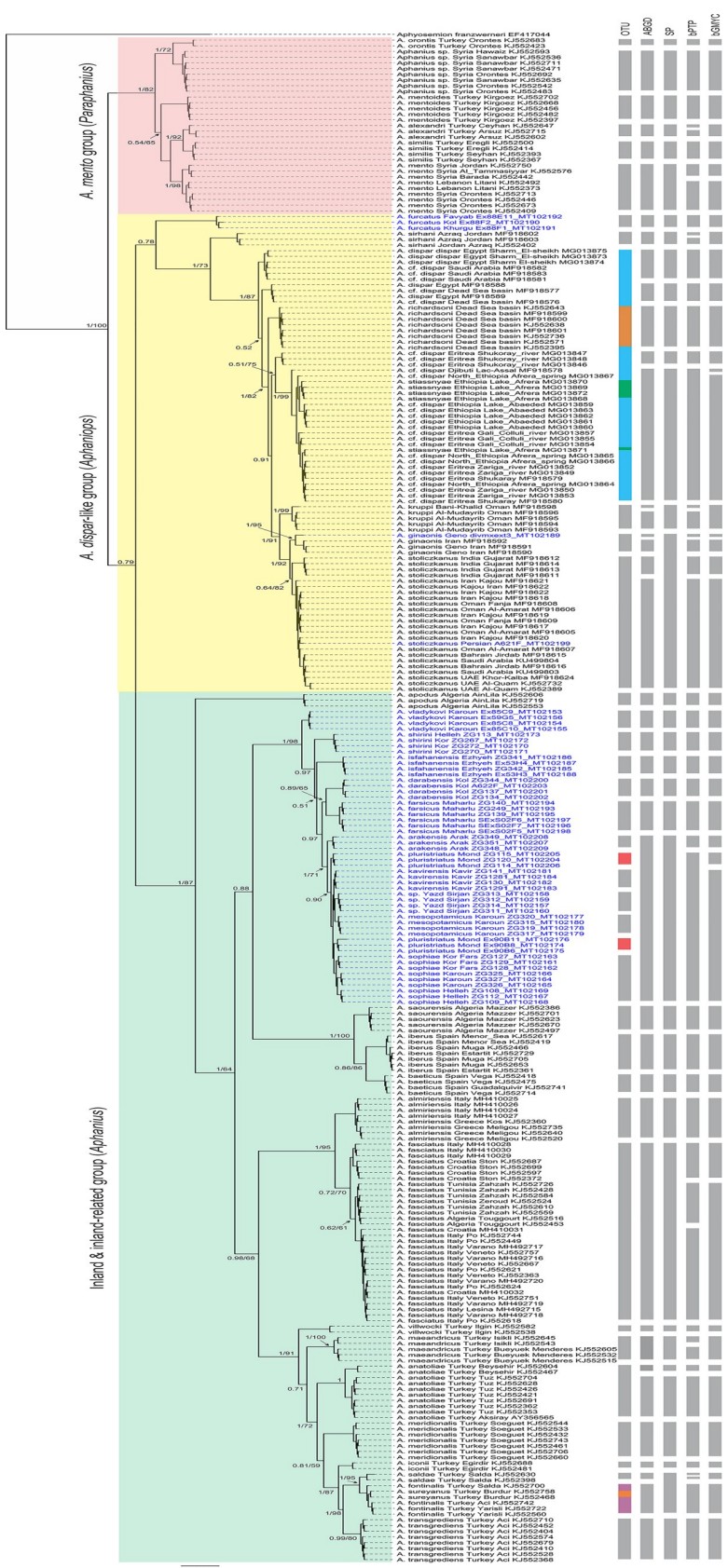

**Fig 2. Bayesian and Maximum Likelihood phylogeny reconstructed based on COI gene sequences visualizing the results of the four different molecular species delimitation Methods.** ABGD, Automatic Barcode Gap Discovery; SP, reversed Statistical Parsimony; bPTP, Bayesian Poisson tree process model; bGMYC, Bayesian General Mixed Yule-Coalescent model. Operational taxonomic unit (OTU) corresponding to the morphological or nominal species. The numbers before and after the slash at important nodes are posterior probability and bootstrap values, respectively (values lower than 0.5 and 50% are not shown at these nodes).

(Eritrea; MF918580 and MF918579); (ii) *A. ginaonis* and *A. stoliczkanus*; (iii) *A. arakensis*, *A. pluristriatus*, *A. kavirensis*, *A. sophiae*, *Aphanius* sp. and *A. mesopotamicus* (all from Iran); and (iv) *A. saldae*, *A. fontinalis*, and *A. sureyanus*.

The bGMYC method split the dataset into 44 species (Fig 2). Out of 40 nominal species, 26 (65%) were delimited through the bGMYC tool. It presented potential species-level diversity in six nominal species: (i) two potential species in *A. mento*; (ii) five potential species in *A. dispar*; (iii) two potential species in *A. ginaonis*; (iv) two potential species in *A. stoliczkanus*; (v) two potential species in *A. fasciatus*; and (vi) two potential species in *A. anatoliae*. The bGMYC method is similar to bPTP, but it uses an ultrametric tree to delimit species. As no mutation rate has been reported for COI in genus *Aphanius*, the maximum age for *Aphanius* was set at 34 Ma based on the oldest known fossil of the Old World killifishes to produce an ultrametric tree. For this reason, and since credible intervals for some species-group nodes were broad (not shown), the bGMYC results should be cautionary interpreted.

Finally, the delimitation method based on bPTP yielded the largest number of species (48 species; Fig 2). Out of 40 nominal species, 21 (52.5%) were delimited through the bPTP tool. It indicated potential species-level diversity in 11 nominal species: (i) two potential species in *A. alexandri*; (ii) two potential species in *A. mento*; (iii) two potential species in *A. sirhani*; (iv) four potential species in *A. dispar*; (v) two potential species in *A. kruppi*; (vi) two potential species in *A. ginaonis*; (vii) two potential species in *A. stoliczkanus*; (viii) three potential species in *A. fasciatus*; (ix) two potential species in *A. maeandricus*; (x) two potential species in *A. anatoliae*; (xi) two potential species in *A. saldae*. This method considered low genetic distance in the COI barcode region for the following groups: (i) *A. stiassnyae* and *A. dispar* specimens from Lac-Assal (Djibouti), Afrera spring (North Ethiopia), Lake Abaeded (Ethiopia), Gali Colluli river (Eritrea), Zariga river (Eritrea), and Shukaray river (Eritrea; including MF918580 and MF918579); (ii) *A. pluristriatus*, *A. kavirensis*, *A. sophiae*, *Aphanius* sp. from Yazd and *A. mesopotamicus*; and (iii) *A. fontinalis* and *A. sureyanus*.

Overall, 28 of 35–48 putative species delimited by the four molecular species delimitation approaches could be morphologically linked to a species name by at least one method, as follows (Fig 2): *A. orontis*, *Aphanius* sp. from Syria, *A. mentoides*, *A. alexandri*, *A. similis*, *A. mento*, *A. furcatus*, *A. sirhani*, *A. richardsoni*, *A. kruppi*, *A. apodus*, *A. vladykovi*, *A. shirini*, *A. isfahanensis*, *A. darabensis*, *A. farsicus*, *A. arakensis*, *A. saourensis*, *A. iberus*, *A. baeticus*, *A. almiriensis*, *A. fasciatus*, *A. villwocki*, *A. maeandricus*, *A. meridionalis*, *A. iconii*, *A. saldae* and *A. transgrediens*. Overall, 10 nominal species represented a high level of diversity by at least one of the four molecular species delimitation methods: *A. alexandri*, *A. mento*, *A. sirhani*, *A. dispar*, *A. kruppi*, *A. ginaonis*, *A. stoliczkanus*, *A. fasciatus*, *A. maeandricus*, and *A. anatoliae*.

In this study, the average K2P genetic distance between a group containing *A. mento* versus a group containing brackish water taxa was 17%, while the average K2P genetic distance between a group containing *A. mento* versus a group containing the inland and inland-related taxa was 19.6% (Table 3). The mean K2P genetic distances within the brackish water group, within the inland and inland-related group, and within the *A. mento* group were 3.7% (S.E. = 0.4%), 13.1% (S.E. = 1%) and 4.8% (S.E. = 0.5%), respectively (Table 4).

**Table 3. Mean K2P distances (%) between three groups of the Old World tooth-carps.**

|  | Inland and inland-related group (*Aphanius*) | *A. dispar*-like group (*Aphaniops*) | *A. mento* group (*Paraphanius*) |
|---|---|---|---|
| Inland and inland-related group (*Aphanius*) | 0 |  |  |
| *A. dispar*-like group (*Aphaniops*) | 19.7 | 0 |  |
| *A. mento* group (*Paraphanius*) | 19.6 | 17.0 | 0 |

### *Paraphanius* gen. nov. Esmaeili, Teimori, Zarei & Sayyadzadeh

(Fig 3)

**Diagnosis.** *Paraphanius* can be distinguished from the other genera of Aphaniidae, *Aphanius* (Fig 4) and *Aphaniops* (Fig 5) by considering the following characteristics: unique colour pattern of breeding males being dark blue-black to dark brown or almost black with iridescent blue-white to silvery spots regularly-arranged on the fins as curved lines, and irregularly on the body (sometimes as irregular vertical bars and sometimes the spots are vertically elongate), the absence of the wide and black bars on the caudal fin of the male (vs. the presence of 3 wide black bars on the caudal fin of the male in *Aphaniops*), the absence of vertical bands on the body of the male, (the presence of vertical bars in the male of *Aphanius*), a higher number of dorsal fin rays than in *Aphaniops* (9–14 in *Paraphanius* vs. 8–9 in *Aphaniops*), a cartilaginous interhyal (versus ossified in *Aphanius* and *Aphaniops*), an embedded urohyal in the fold of branchiostegal membranes (versus not embedded in *Aphanius* and *Aphaniops*), an upturned lower jaw (versus not upturned in *Aphanius* and *Aphaniops*), and a distinctive neuromast pattern on the dorsal surface of the head (versus less prominently developed in *Aphanius* and *Aphaniops*) based on Parenti [28]. The epural is thin or less developed in *Paraphanius*, while it is thick and well developed in *Aphanius* and *Aphaniops*. Similar to *Aphanius*, the otolith in *Paraphanius* has a straight sulcus (versus a sulcus which is distinctly bent terminally in *Aphaniops*).

**Remark.** It has been proposed that the *Paraphanius* group is more closely related to the Anatolian *Kosswigichthys* Sözer, 1942 and the South American *Orestias* Valenciennes in Cuvier and Valenciennes, 1846 than the genus *Aphanius* (Parenti [28]:524). However, *Kosswigichthy* is considered as a synonym of *Aphanius* Nardo, 1827 by Wildekamp [16]:19. In addition, according to osteological information provided by Parenti [28]:521, Aphaniidae is distinguished from Orestiidae by having a vomer (vs. absent), a cartilaginous mesethmoid (vs. ossified) and an ossified interhyal (not ossified). Based on the tree topology, populations from the Orontes (= Asi) and the Levant in coastal and Dead Sea basins should be considered as distinct taxa.

Based on the tree topology, *A. alexandri*, *A. mento*, *A. mentoides*, *A. orontis*, *A. similis*, and *Aphanius* sp. make a distinct clade sister to *A. dispar* plus *A. fasciatus* group. The validity of the *A. mento* group has been under debate. Akşiray [109] published the description of one new species and 15 new subspecies of *Aphanius* from Turkey. However, Villwock [110, 111] as well

**Table 4. Mean of the within group K2P distance (%) for the Old World tooth-carps.**

| Tooth-carp groups | Mean of the within group K2P distance | S.E.[a] |
|---|---|---|
| Inland and inland-related group (*Aphanius*) | 13.1 | 1 |
| *A. dispar*-like group (*Aphaniops*) | 3.7 | 0.4 |
| *A. mento* group (*Paraphanius*) | 4.8 | 0.5 |

[a]standard error

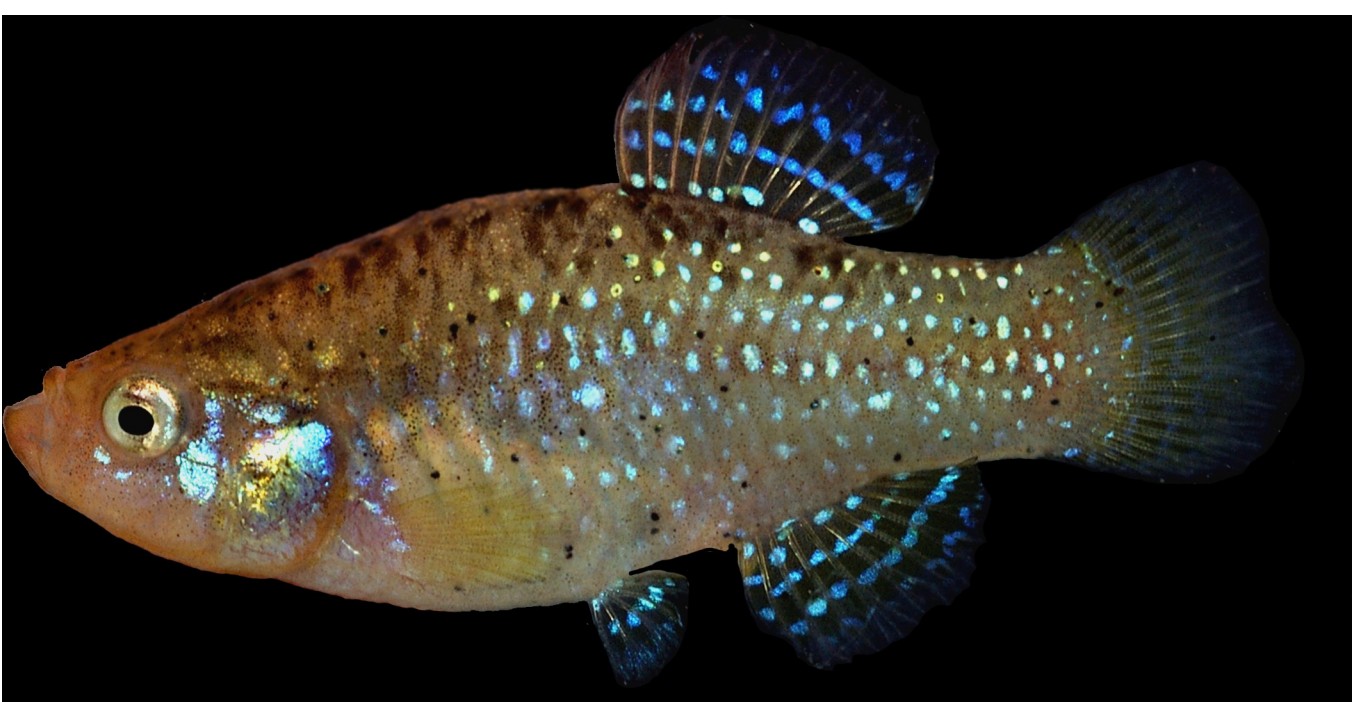

**Fig 3. *Paraphanius mento*, male, Spring of Barada (provided by J. Freyhof).**

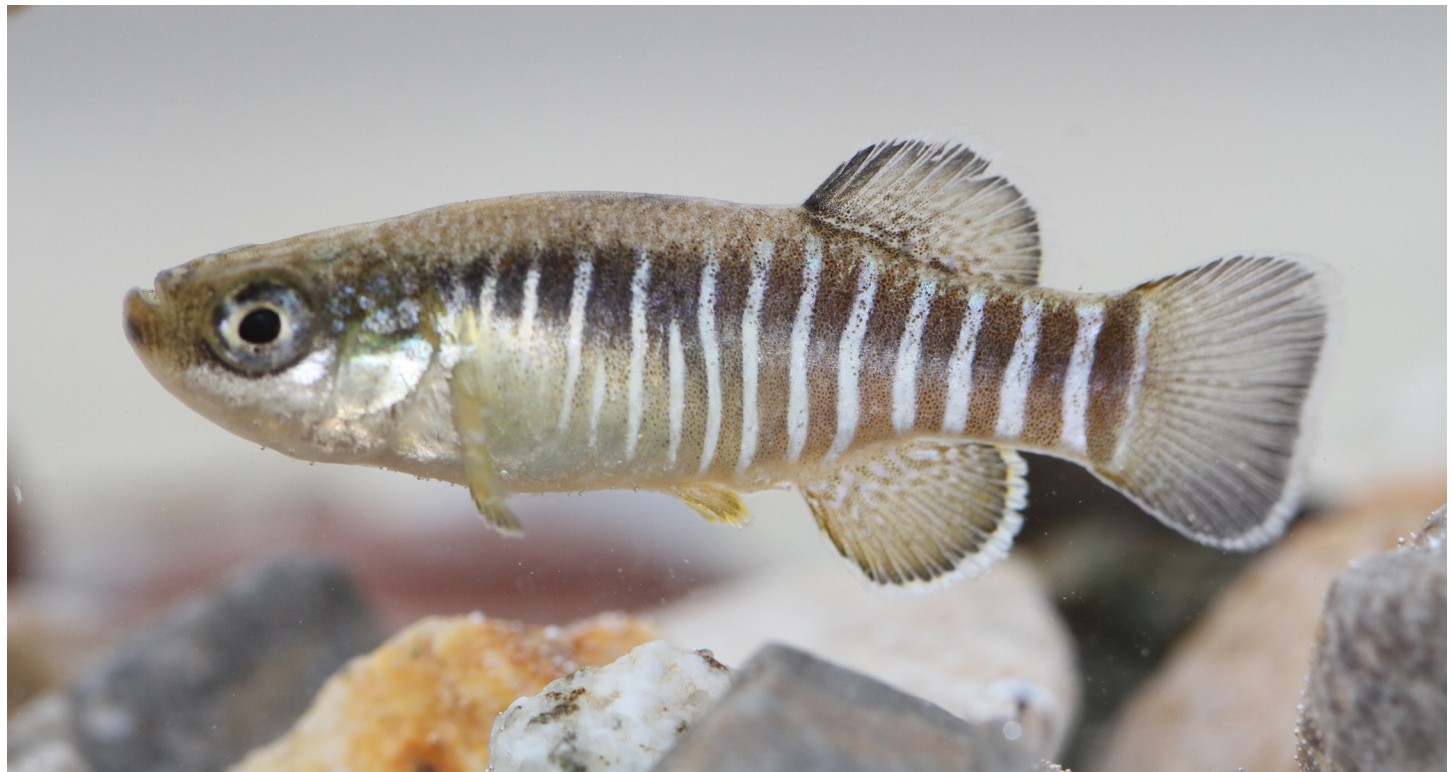

**Fig 4. *Aphanius pluristriatus*, male, Mond River drainage, Iran.**

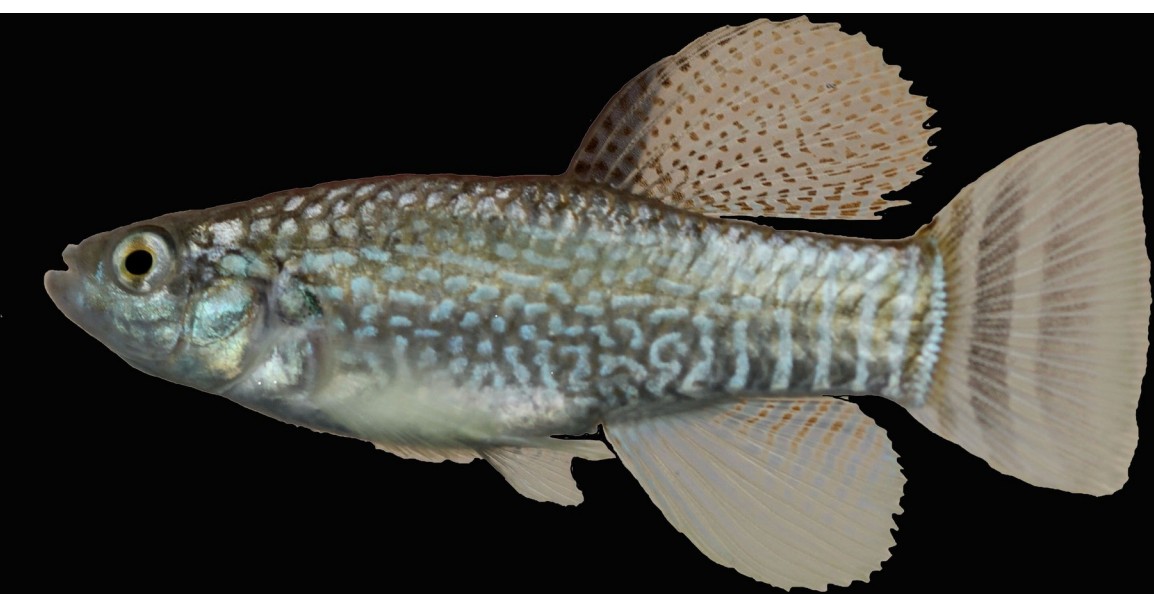

**Fig 5. *Aphaniops stoliczkanus*, male, Tigris River drainage, Ilam, Iran.**

as Wildekamp [16] and Wildekamp et al. [23] did not accept any of these 16 taxa as valid. Wildekamp et al. [23] placed *A. cypris alexandri*, *A. cypris boulengeri*, *A. cypris orontis*, *A. sophiae similis* and *A. sophiae mentoides* in the synonymy of *A. mento*. These subspecies have species rank and are considered in the genus *Paraphanius* here.

The species diagnosis is valid also as a genus-level diagnosis.

**Type species.** *Aphanius mento* (Heckel, 1843), Mossul, northern Iraq (36˚18'N, 43˚18'E). Possible syntypes: NMW 21699–704 (6), 59832 (21), illustrated by Heckel 1843: Pl. 6 (Fig 4).

**Etymology.** The generic name, *Paraphanius*, is made out of the Greek word Para, meaning "near, resembling", and *Aphanius*, from the Greek word Aphanus, meaning "invisible", a genus of tooth-carp fish, in reference to its phylogenetic position being sister to *Aphanius* and *Aphaniops*.

Found in the Orontes (= Asi) and Tigris-Euphrates basins, the Levant in coastal and the Dead Sea basins, western Jordan, and in southern Turkey in Mediterranean basins as well as in central Turkey.

***Paraphanius mento*, the type species.** Scales along the body 23–28, a total of 9–14 rays in the dorsal fin, a total of 9–13 in the anal fin, 12–16 and 4–6 rays in pectoral and pelvic fins respectively. Total gill rakers 11–15. The jaw teeth display a long middle cusp and short lateral cusps, otoliths are triangular in shape, have a prominent rostrum, and a straight sulcus. A total of 25–26 vertebrae present in the vertebral column, five preural vertebrae (PU1–5) with long haemal and neural spines, the presence of a small plate in front of the spines of PU5, flank scales are squarish to a vertical oval in general shape. There is no pelvic axillary scale. The gut is s-shaped.

**Included species.** *Aphanius alexandri*, *A. mento*, *A. mentoides*, *A. orontis*, *A. similis*, and *Aphanius* sp.

## Discussion

### Species delimitation

Traditional morphology-based approaches are unable to distinguish cryptic entities and unveil the loss of genetic diversity. Hence, they underestimate the true biological diversity, and

protecting only the visible biodiversity adds additional level of uncertainty for monitoring and management actions of biodiversity and evolutionary processes [112–115]. Therefore, recent taxonomic and phylogenetic studies are mostly based on morphological and molecular data [116–125]. In this study, we considered the phylogenetic concept of the species and discussed also the previously published morphological characteristics of the species. It should be also noted that in this section, we refer to the name of three genera *Aphanius*, *Aphaniops*, and *Paraphanius* for referring their representative species. By applying a COI barcode library for the tooth-carps fishes of the Old World, we conducted an array of molecular species delimitation analyses (i.e., ABGD [100], SP [102], bPTP [104], and bGMYC [105]) to investigate species boundaries and to compared the results to existing morphology-based taxonomy. Across the entire tree, the number of species-level MOTUs (i.e. molecular operational taxonomic units) varied depending on the technique applied, ranging from 35 in SP to 48 in bPTP. The combination of these methods identified 28 (70%) among 40 nominal species. Several MOTUs did not correspond to nominal species, revealing potential cases of synonymy and the presence of species-level cryptic diversity within nominal species. Nevertheless, all the known tooth-carp species were monophyletic except *Aphaniops dispar*, *Aphaniops stiassnyae*, *Aphanius pluristriatus*, and *Aphanius fontinalis*. The majority of methods used split several nominal species including *Aphaniops dispar*, *Aphaniops stoliczkanus*, *Aphanius fasciatus*, and *Aphanius anatoliae* into the independent lineages, which might reflect deep divergences occurring between populations across these species distributions. The species delimitation methods used here revealed a low diversity of the COI barcode region in several species: (i) *Aphanius fontinalis* with *Aphanius sureyanus*; (ii) *Aphaniops stiassnyae* with *Aphaniops dispar*; and (iii) *Aphanius pluristriatus*, *Aphanius kavirensis*, *Aphanius* sp. from Yazd and *Aphanius mesopotamicus* within the *Aphanius sophiae* group. The complexity of the *A. sophiae* species group has already been discussed in detail [19]. According to Esmaeili et al. [19], the *A. sophiae* subclade includes the most diverse subclade of inland and inland-related species (IIRAS). These authors subdivided the *A. sophiae* subclade into three lineages according to their temporal diversification including i) the lineage of *Aphanius isfahanensis* containing only this species, which diverged much earlier (4.8. m.y.a. according to Hrbek et al. [40]) than all other species of the *A. sophiae* subclade, ii) *Aphanius farsicus* + *Aphanius arakensis* which probably diverged in the Late Pleistocene (100,000–11,700 y. ago), and iii) the lineage of *A. mesopotamicus* comprising a group of very closely related species (*A. sophiae*, *A. mesopotamicus*, *A. pluristriatus*, *A. kavirensis*) that may have diversified 11,700 to 4,000 y. ago (Early to Middle Holocene) (see also Gholami et al. [48]). Notably, external characters do not unambiguously distinguish between the species of the *A. sophiae* subclade, with the exception of *A. isfahanensis*. However, differences in cytochrome *b* sequences and also between the otoliths clearly show that the species of the *A. sophiae* subclade are distinct [47–48]. In addition, they inhabit widely separated distribution areas without any hydrological networks or connectivity between, which promotes speciation (Fig 1). These are the reasons that we consider the species of the *A. sophiae* subclade as distinct species rather than as populations of a single species.

Moreover, Geiger et al. [86] and Behrens-Chapuis et al. [126] showed that the COI barcode region could not resolve a notable number of fish species. This situation is also usually found in other animal groups [3, 127–129]. The disagreement between distances in the COI barcoding gene sequence and morphologically recognized species might be due to introgressive hybridization, fast evolution, recent speciation events (young group) and genetic drift [86, 126, 130].

Applying molecular species delimitation methods revealed high genetic diversity in *Aphaniops ginaonis*, a species restricted to Genow hot spring in the Hormuzgan basin, southern Iran. This spring is geographically very close to another endemic tooth-carp in this basin,

*Aphaniops hormuzensis*. Variation in the population of *A. ginaonis* has already been noticed and discussed by considering otolith morphology [131]. In the study of Reichenbacher et al. [131], some otoliths showed an atypical morphology i.e., a reduced rostrum length and a lower or higher length/height value. The presence of these otoliths highlighted the possibility of hybridization between *A. ginaonis* and *A. hormuzensis*. Here, such variability is observed in the COI barcode region of studied *A. ginaonis* individuals. Since this spring is currently used by local people for hydrotherapy, possible hybridization cannot be ruled out in this hot spring.

Systematics and biogeography of the *Aphaniops dispar* group have already been discussed by Teimori et al. [15]. The most common species in the family Aphaniidae is *A. dispar* and it has long been regarded as a species group rather than a single species. Based on the data provided by Teimori et al. [15] and Freyhof et al. [10], nine species are recognized in the *A. dispar* group. The "true" *A. dispar* from the Red and the Mediterranean Sea basins, *Aphaniops stoliczkanus* from the coastal areas of the Persian Gulf, the northern Arabian Sea east to Gujarat in India, the Gulf of Oman and some endorheic basins in Iran and Pakistan, *Aphaniops richardsoni* from spring-stream systems in the Dead Sea basin, *Aphaniops sirhani* from the Azraq Oasis in Jordan, *A. ginaonis* from a single spring in Hormuzgan basin S-Iran, *Aphaniops furcatus* from a few streams and springs in Hormuzgan basin S- Iran, *A. hormuzensis* from Hormuzgan basin S- Iran, *A. stiassnyae* from one lake in Ethiopia and *Aphaniops kruppi* from the Wadi Al Batha drainage in northern Oman. Hence, based on the Teimori et al. [15], Freyhof et al. [10], and the data presented here, especially molecular species delimitation methods using the COI barcode region (Fig 2) for *Aphaniops dispar*, the hypothesis of the presence of a single widespread coastal species in the Middle East is rejected and makes it possible that more additional unidentified species occur in the Red Sea basin.

Since biodiversity is in decline due to climate change and anthropogenic land use [132, 133], species are vanishing at a distressing rate while a growing body of evidence underpins the considerable impact of biodiversity loss on the functioning of ecosystems [134–136]. This is also true in the case of the studied fishes, which are the only members of the tooth-carps in the Old World [16], and most of their species are endemic to the landlocked water systems and are under threat owing to natural and human-induced disturbances [14]. These disturbances even forced some species, such as *Aphanius farsicus*, to the edge of extinction [137]. In this context, the efficiency of species identification in some aphaniids was demonstrated in the present study by DNA barcoding and emphasized that COI sequencing could be used to reconstruct the evolutionary relationships within the family Aphaniidae.

## Taxonomic remarks on the genera *Aphanius*, *Aphaniops*, and *Paraphanius*

As suggested by Hrbek & Meyer [11], the genus *Aphanius* could be ecologically divided into two main groups: The first group contains those species inhabiting brackish water and euryhaline coastal environments such as the members of *A. dispar* group. The second group includes those species inhabiting freshwater systems such as spring-streams, creeks, marshes, and lakes in landlocked basins. Good examples for the latter are those species inhabiting the inland water systems in Anatolia and Iran. The molecular-based study of Hrbek & Meyer [11], highlighted two major phylogenetic clades for the Palearctic killifishes that correspond to the former eastern and western Tethys Sea. Within the eastern clade, two distinct groups have been recognized. One was the freshwater group inhabiting the Arabian Peninsula and the second one was an euryhaline group inhabiting the coastal regions from Pakistan to Somalia. Within the western clade, they recognized a group containing the killifishes of the Iberian Peninsula and the Atlas Mountains, and Turkey and Iran. Based on the tree topology presented here, the *Paraphanius mento* group (*P. alexandri*, *P. mento*, *P. mentoides*, *P. orontis*, *P. similis*,

and *Paraphanius* sp.) make a distinct clade sister to *Aphaniops dispar* plus the inland and inland-related tooth-carps/*A. fasciatus* groups. Parenti [28], has been summarized several diagnostic features for *Aphanius mento* (now *Paraphanius mento*) that are not found in the members of *Aphanius* and *Aphaniops*, including a cartilaginous interhyal (ossified in *Aphanius* and *Aphaniops*), an embedded urohyal (not embedded in *Aphanius* and *Aphaniops*, see also Teimori et al. [59]), an upturned lower jaw (not upturned in other *Aphanius* and *Aphaniops*), and a distinctive neuromast pattern on the dorsal surface of the head (less prominently developed in *Aphanius* and *Aphaniops* species). Therefore, she proposed *A. mento* as a derived member of the *Aphanius*-like clade that should be designated as "*Aphanius*". This taxonomic separation has later been supported by the observation of the sulcus morphology of the otoliths (Reichenbacher et al. [29], see below).

A study by Reichenbacher et al. [29] used the shape of the otolith sulcus and provided a parameter for the preliminary assignment of the Old World cyprinodontiforms to distinct groups. Based on their study, the Old World tooth-carps can be assigned into two groups: Group I consists of the otoliths with straight sulcus (Fig 4 in Reichenbacher et al. [29]) and includes the Mediterranean species *Aphanius baeticus*, *A. iberus*, *A. fasciatus*, *A. mento* (now *Paraphanius mento*), and the Turkish species; Group II consists of the otoliths with a sulcus distinctly bent terminally (Fig 7 in Reichenbacher et al. [29]). The latter group includes the tooth-carps from the Arabian Peninsula, *A. ginaonis* (now *Aphaniops ginaonis*), and *A. sirhani* (now *Aphaniops sirhani*).

Hrbek & Meyer [11] sorted one species from Group I, i.e., *Aphanius mento* (now *Paraphanius mento*), into the brackish water group consisting of *A. dispar*, *A. ginaonis*, and *A. sirhani*, which represent the Group II in the study of Reichenbacher et al. [29] (see above). As a result, Reichenbacher et al. [29] considered *A. mento* to be an intermediate form between the two groups or clades. In the present study, the phylogenetic analyses revealed that a clade containing *Paraphanius mento* (plus *P. orontis*, *P. mentoides*, *P. alexandri*, *A. similis*, *Paraphanius* sp.), has a basal position in the phylogenetic tree of the Old World tooth-carps, and it is sister to the brackish water *A. dispar* group plus the clade containing inland and inland-related tooth-carps/*A. fasciatus* group. This phylogenetic separation is not completely consistent with the molecular phylogeny provided by Hrbek & Meyer [11], and also partly with the outcomes of Reichenbacher et al. [29].

In addition to the *P. mento* clade, in our DNA barcoding analysis on the Old World tooth-carps, two other large groups were recognized. The first one was the *A. dispar*-like brackish water tooth-carps containing also *A. furcatus* (a brackish water scaleless tooth-carp) and *A. sirhani*. The second group contains all the inland and inland related tooth-carps plus *A. fasciatus* and *A. apodus*. It should be noted that all the inland tooth-carps from Anatolia, Turkey, and Iran fall in the second group, and each of them formed a monophyletic group. This pattern of divergence has proved the hypothesis that all of the Old World tooth-carps originated from an ancestor populated the coastal environments.

Here, we listed several osteological features that show differences between these two large groups (i.e., the members of genus *Aphaniops* vs. those members of the genus *Aphanius*). (i) The number of total vertebrae in *Aphaniops* varies from 25 to 28, verses 26 to 29 in *Aphanius*, (ii) the vertebral columns in *Aphaniops* are often straight versus often slightly curved in *Aphanius*, (iii) the total number of principle caudal-fin rays (PCFR) varies from 9 to 12 (often 10) in *Aphaniops* versus 10–12 (often 11) in *Aphanius*, (iv) the proximal limit of the parhypural in *Aphaniops* is placed very close or has a moderate distance to preural centrum 1, while it has often a clear distance to preural centrum 1 in *Aphanius*, (v) the presence of a constriction on the proximal portion of the neural spine of the preural centrum 2–4 in *Aphaniops*, whereas no obvious constriction presents on the proximal portion of the hemal spine of the preural

centrum in *Aphanius*, (vi) the neural and hemal spines of preural centrum 4 supports at least some caudal fin rays in *Aphaniops*, while they do not reach and support the caudal fin rays in *Aphanius*.

The taxonomic status based on morphology shows the presence of three distinct genera in the family Aphaniidae is confirmed with genetic distances. The average genetic distance among the populations and subspecies is 0.0137% and 0.171%, respectively [138]. Among the genera, this value ranges from 7.70% to 30.50% (mean 23.70%), and within the family, it ranges from 17% to 31% (mean 25%) [138]. In this study, the average K2P genetic distance between a group containing *P. mento* versus a group containing brackish water taxa was 17%, while the average K2P genetic distance between a group containing *P. mento* versus a group containing the inland and inland-related taxa was 19.6% (Table 3). The mean genetic K2P distance within the brackish water group, within the inland and inland-related group, and within the *P. mento* group were 3.7% (S.E. = 0.4%), 13.1% (S.E. = 1%) and 4.8% (S.E. = 0.5%), respectively (Table 4).

Based on the literature review given in the manuscript, morphological comparisons among the distinct groups of the Old World tooth-carps, and species delimitation approached examined in this study, we recognized the genera *Aphanius* Nardo, 1827 and *Aphaniops* Hoedeman, 1951 as valid. The type species for the genus *Aphanius* Nardo, 1827 is *Aphanius nanus* Nardo, 1827 (= *A. fasciatus*) by subsequent designation of Jordan [25]. The type species for the genus *Aphaniops* is *Lebias dispar* Rüppel, 1829 [111].

The genus *Aphanius* contains all the inland and inland-related tooth-carps, which are mainly distributed in the inland waters in Turkey and Iran and also in the inland-related environments around the Mediterranean basin (Fig 2). The genus *Aphaniops* contains all the brackish water in the coastal lines around the Red Sea and the Persian Gulf basins (Fig 2).

Parenti [28] used the *Aphanius mento*-complex to include *mento* (Heckel) and *chantrei* (Gaillard, 1895). Also, Parenti [28] placed the genus name *Aphanius* in parentheses because she noted several deviations from "normal" *Aphanius* species concerning the following characteristics: urohyal embedded in the fold of the branchiostegal membranes and a derived head region pore pattern. As mentioned above, some otolith features have also been reported by Reichenbacher et al. [29] to be unique in *A. mento* and not find in other species of the genus *Aphanius*. Thus, they proposed *A. mento* to be an intermediate form between the two *Aphanius* groups (*Aphanius* and *Aphaniops* in this study).

Parenti [28, 139] places this species in the genus "*Aphanius*", i.e. distinct from true *Aphanius* without defining and naming a new genus. This genus is more closely related to the Anatolian *Kosswigichthys* Sözer, 1942 and the South American *Orestias* Valenciennes in Cuvier and Valenciennes, 1846 than true *Aphanius*.

**Comparative material.**   See S1 Appendix for the list of comparative material examined. The list includes those materials examined by Esmaeili et al. ([50]: *farsicus*), Esmaeili et al. ([19]: *A. darabensis* and *A. kavirensis*); Teimori et al. ([47]: *A. arakensis*), Gholami et al. ([48]: *A. shirini*); Gholami et al. ([20, 80]: *A. farsicus* and *A. sophiae*), Teimori et al. ([51]: *A. furcatus*) and Teimori et al. ([15]: *A. dispar* group).

**Nomenclatural acts.**   The electronic edition of this article conforms to the requirements of the amended International Code of Zoological Nomenclature, and hence the new names contained herein are available under that Code from the electronic edition of this article. This published work and the nomenclatural acts it contains have been registered in ZooBank, the online registration system for the ICZN. The ZooBank LSIDs (Life Science Identifiers) can be resolved and the associated information viewed through any standard web browser by appending the LSID to the prefix "http://zoobank.org/". The LSID for this publication is: urn:lsid:zoobank.org:pub:A8F5EA21-50CE-4FBE-969C-EA6B61FFBABE

## Supporting information

**S1 Appendix. List of comparative material examined.**
(DOCX)

**S1 Table. Mean K2P distances between the studied tooth-carps within the family Aphaniidae.**
(XLS)

## Acknowledgments

We are pleased to thank U. Schliewen and D. Neumann, Department of Ichthyology, Bavarian Natural History Collections, SNSB Bavarian State Collection of Zoology (ZSM) for providing X-ray photography facilities. We thank Z. Gholami helping with DNA extraction. We thank R. Zamanian Nejad, S. Babaee, R. Khaefi, M. Masoudi, and H.R. Mehraban helping with fish collection. We thank B.W. Coad for editing the ms. The research work was approved by the Ethics Committee of Biology Department (SU-9233856).

## Author Contributions

**Conceptualization:** Hamid Reza Esmaeili, Azad Teimori.

**Data curation:** Hamid Reza Esmaeili, Azad Teimori, Fatah Zarei, Golnaz Sayyadzadeh.

**Formal analysis:** Hamid Reza Esmaeili, Azad Teimori, Fatah Zarei, Golnaz Sayyadzadeh.

**Funding acquisition:** Hamid Reza Esmaeili.

**Investigation:** Hamid Reza Esmaeili, Azad Teimori.

**Methodology:** Hamid Reza Esmaeili, Azad Teimori, Fatah Zarei, Golnaz Sayyadzadeh.

**Software:** Hamid Reza Esmaeili, Fatah Zarei.

**Supervision:** Hamid Reza Esmaeili.

**Writing – original draft:** Hamid Reza Esmaeili, Fatah Zarei, Golnaz Sayyadzadeh.

**Writing – review & editing:** Hamid Reza Esmaeili, Azad Teimori, Fatah Zarei, Golnaz Sayyadzadeh.

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
