## [Decision Letter · Decision Letter 0]

13 Feb 2020

PONE-D-19-33131

DNA barcoding and species delimitation of the Old World tooth-carps, family Aphaniidae Hoedeman, 1949 (Teleostei: Cyprinodontiformes)

PLOS ONE

Dear Prof. Esmaeili,

Thank you for submitting your manuscript to PLOS ONE. After careful consideration, we feel that it has merit but does not fully meet PLOS ONE’s publication criteria as it currently stands. Therefore, we invite you to submit a revised version of the manuscript that addresses the points raised during the review process. 

We would appreciate receiving your revised manuscript by Mar 29 2020 11:59PM. To enhance the reproducibility of your results, we recommend that if applicable you deposit your laboratory protocols in protocols.io, where a protocol can be assigned its own identifier (DOI) such that it can be cited independently in the future. For instructions see: http://journals.plos.org/plosone/s/submission-guidelines#loc-laboratory-protocols

We look forward to receiving your revised manuscript.

Kind regards,

Roberta Cimmaruta, PhD

Academic Editor

PLOS ONE

Journal Requirements:

2. In your Methods section, please provide additional location information of the sampling sites, including geographic coordinates for the data set if available.

3. To comply with PLOS ONE submissions requirements, please provide methods of sacrifice in the Methods section of your manuscript.

4. We noticed you still have some occurrence of overlapping text with the following previous publications, which needs to be addressed:

https://link.springer.com/article/10.1007%2Fs10641-009-9549-5

https://onlinelibrary.wiley.com/doi/full/10.1046/j.1420-9101.2003.00475.x

https://www.biotaxa.org/Zootaxa/article/view/zootaxa.4338.3.10

In your revision ensure you cite all your sources (including your own works), and quote or rephrase any duplicated text outside the methods section. Further consideration is dependent on these concerns being addressed.

5. We note that Figure 1 in your submission contains map images which may be copyrighted. All PLOS content is published under the Creative Commons Attribution License (CC BY 4.0), which means that the manuscript, images, and Supporting Information files will be freely available online, and any third party is permitted to access, download, copy, distribute, and use these materials in any way, even commercially, with proper attribution. For these reasons, we cannot publish previously copyrighted maps or satellite images created using proprietary data, such as Google software (Google Maps, Street View, and Earth). For more information, see our copyright guidelines: http://journals.plos.org/plosone/s/licenses-and-copyright.

You may seek permission from the original copyright holder of Figure 1 to publish the content specifically under the CC BY 4.0 license. 

If you are unable to obtain permission from the original copyright holder to publish these figures under the CC BY 4.0 license or if the copyright holder’s requirements are incompatible with the CC BY 4.0 license, please either i) remove the figure or ii) supply a replacement figure that complies with the CC BY 4.0 license. Please check copyright information on all replacement figures and update the figure caption with source information. If applicable, please specify in the figure caption text when a figure is similar but not identical to the original image and is therefore for illustrative purposes only.

Reviewers' comments:

Reviewer's Responses to Questions

**Comments to the Author**

1. Is the manuscript technically sound, and do the data support the conclusions?

Reviewer #1: Yes

Reviewer #2: Partly

2. Has the statistical analysis been performed appropriately and rigorously? 

Reviewer #1: Yes

Reviewer #2: No

3. Have the authors made all data underlying the findings in their manuscript fully available?

Reviewer #1: Yes

Reviewer #2: No

4. Is the manuscript presented in an intelligible fashion and written in standard English?

Reviewer #1: No

Reviewer #2: No

5. Review Comments to the Author

Reviewer #1: Dear Editor,

the manuscript PONE-D-19-33131 “DNA barcoding and species delimitation of the Old World tooth-carps, family Aphaniidae Hoedeman, 1949 (Teleostei: Cyprinodontiformes)” is an interesting paper dealing with the phylogenetic status of the peri-mediterranean genus Aphanius. The authors splitted up the species of this genus in the new genus Paraphanius, the genus Aphaniops (Hoedeman, 1951) and Aphanius Nardo, 1827, by applying four different molecular species delimitation methods to a consistent molecular dataset of cytochrome c oxidase I sequences. The methodology is appropriate and support the results and the main conclusion of the manuscript. The cited literature is quantitatively important and exhaustive and testifies to the authors' expertise. In my opinion this manuscript is of interest for the publication on Plos, but not in the present form because it needs major revision. First of all the text should be revised by a native English speaker. I found the text hard to read in many parts with sentences that could be written more clearly. However, I would like to stress that the discussion is much more fluent to read than the rest of the manuscript. The authors could lighten the text making it more effective for reading in some parts.

Line 27 relic should be relict

Line 39 “the family, including (i)…” “the family: (i) the first clade, positioned at the base of the phylogenetic tree, include A. mento and…..”

Line 42 “third cladecontains” “ third clade contains”

Lines 55-56 “In this concept, morphology-based methods have high efficiency.” This sentence could be deleted

Line 57 “often lack the ability to discover the hidden species,” could be written “often does not allow to discover criptic species”

Line 62 “DNA barcoding with…” “DNA barcoding based on…”

Line 62 “ …cytochrome c oxidase I), COI(“ should be “…cytochrome c oxidase I (COI)”

Line 64 “…with a low average distance within species of 0.39% [4].” This sentence could be changed “…only if the average intraspecific and interspecific genetic distance are significantly different (barcoding gap).

Lines 65-67 “DNA barcoding has been sufficiently used for the species identification because of the universal primers described by Ward et al. [4] and Ivanova et al. [5] that were very effective for the amplification of the COI sequences of most species” This sentece could be deleted.

Line 68 Please delete however

Line 68 “acuracyto” “accuracy to”

Line 74 “along with” should be “along”

Line 76 “..have largely been affected their..” should be “have largely affected their..”

Line 76 “ distribution. .” delete the point

Lines 77-81 this sentence must be written in a correct English!

Lines 89-90 “Aphanius princeps is the taxon with the oldest fossil skeleton, and found in the deposits of Burdigalian age in Catalonia, NE Spain [8].” could be “Aphanius princeps, found in the deposits of Burdigalian age in Catalonia, NE Spain, is the taxon with the oldest fossil skeleton [8].”

Lines 91-95 Please simplify this sentence “The genus Aphanius is the only native representatives, in the Old World, of the family Cyprinodontidae recently renamed Aphaniidae [9, 10-12], as firstly suggested by Sethi in 1960 [13]"

Line 111 Please include the reference to Lazara.

Line 113 “La Cepede” should be “Lacépède”

Line 117 “…however Aphanius dispar….” Should be “…while Aphanius dispar…”

Lines 119- 122 and 123-128 These two sentences are repetitive. The authors should keep one of them.

Line 138-175 The information of this paragraph could be summarized on a table.

Line 180 “..represents..” should be “..host..”

Line 186 “…up until recently..” could be deleted

Line 193”..delamination..” should be “delimitation”

Line 194 Delete “for”

Lines 195-198 “For this……of these fishes” Delete this ripetitive sentence

Line 288 Delete “was”

Line 301, 319 and 325“….delimited obviously..” Line 310 “..delimited nicely..” In don’t understand why the authors use these adverbs.

Lines 332 “.. genetic distance in the COI barcode region the following groups..” should be “.. genetic distance in the COI barcode region for the following groups..”

Lines 412-413 Delete “..also lack the ability..”

Lines 416-417 “Therefore recent taxonomic and phylogenetic studies are based on morphological and molecular data”

Line 420 “..and compared the results..” should be “ ..and to compare the results..”

Line 436 “according to the temporal diversification…” “ according to their temporal diversification.”

Line 436 “..A. isfahanensis contains..” “..A. isfahanensis containing..”

Line 438 “..probably diverged..” “..which probably diverged..”

Line 445 “..subclade are distinct” : please add a reference

Line 461 “…length height value..” …lenght/height va

Reviewer #2: DNA barcoding and species delimitation of the Old World tooth-carps, family

Aphaniidae Hoedeman, 1949 (Teleostei: Cyprinodontiformes), Esmaeili et al.

This is an interesting manuscript that uses COI barcoding as a method for species delimitation within the Aphaniidae and proposes a new genus, using both published and new molecular data. The objectives of this ms. are 1) to use DNA barcoding as a tool for species delimitation within the family Aphaniidae; 2) to give a clearer picture of intra-family relationships; and 3) to clarify the generic names for the species groups within the family.

As presented, this manuscript needs to improve a few issues before can be accepted for publication.

First of all, only 14 species were identified (lines 212-213) and sequenced by the authors, while the others sequences belong to published information in part by the same group of authors. Unfortunately, not always the species identification of sequences deposited in Genbank is correct, and this might explain partially the disagreement between current taxonomy and the phylogenetic analysis. I suggest to clarify/explain the identification of species done by the authors and based in morphological diagnostic characters. If diagnostic characters are not external, then species identification might not be reliable (as specimens need to be cleared and stained for example). If some individuals were not identified, either by the authors or others, then these individuals should be clearly marked/identified. Another suggestion is to separate the analysis in two: only considering information gathered by the authors (i.e. information that the authors can trust), and a second analysis including published sequences.

Both species identification (lines 212-213) and molecular methods (220-223) are too brief. If the authors do not want to include full methods within the text please add a supplement including details of morphological characters for species identification and methods that clarify molecular tasks. The main results of this ms. are based in these methods, so they should be included in this ms. and not citing third party or other papers from the authors. It is really important to explain the readers how the individuals were identified and how the sequences were obtained.

The COI gene might not be a good marker for phylogenetic analysis, as it might be saturated. Please include an analysis showing that the analysis is not skewed due to saturation (e.g. using DAMBE [Xia & Xie 2001, J Hered 92(4):371-3] or other software). If this is the case, then the phylogenetic analysis cannot be presented as is, and the authors should refer to it as a phenetic analysis. In this regard, the outgroup used might not be adequate, please include other species/genera as there is no mention for the systematic account of the Aphaniidae and relatives. It should be noted that the new sequences are not deposited in Genbank yet, and thus are not accessible (line 234).

Considering the fossil calibration (lines 245-247), the results should be cautionary interpreted. Have the authors calculated a mutation rate? It seems that it is much higher than “standard” rates of 1-3% per million year. If this is the case, then the ultrametric tree produced by BEAST might be biased and I suggest not to use it (or to change/delete the fossil date for calibration). Also, BEAST produce a credible interval for each node: are these broad? These re-analyses might change completely the molecular results and interpretation. It should be noted that the K2P distances are used and not phylogenetic corrected distances (lines 346-351). These are phenetic but not phylogenetic results. Also, please see Srivathsan & Meier (2012). On the inappropriate use of Kimura-2-parameter (K2P) divergences in the DNA-barcoding literature. Cladistics 28, 190–194.

Figure 2 include two numbers at some nodes (posterior probabilities and bootstrap values) but not for all of them. Why not every single node has these 2 values? This mean that a node was not recovered with both methodologies? Please discuss the validity of the support at each node, especially if some nodes were not recovered with both approaches.

One of the strengths of this ms. is that the authors seek for an agreement between morphological characters and a molecular approach which is one of the main contributions of the ms. I suggest to include a table where the reader can easily understand if each species has a valid diagnosis (whether the characters are internal or external) and if according to the molecular methodology also would be a valid species. Sometimes the results are not clear enough and this addition might improve both the quality and clarity of the ms. Please also discuss (and add to the methodology) which species concept each of the 4 molecular methods stand for (see De Queiroz, Syst. Biol. 56, 879–886 [2007]). Under some species concepts (e.g. monophyly) one species might not be valid but can be diagnosable. This highlight the fact that some morphological characters are based in males and COI gene has maternal inheritance, please discuss this if that is occurring.

Regarding morphological characters and specifically the systematics of the 3 genera proposed under Aphaniidae, it is not completely clear which characters are diagnostic, and which belong to the description of each genus (lines 362-363: characters overlap between genera, e.g. dorsal fin rays; and are not described for Aphanius; lines 543-551: number of total vertebrae, principle caudal-fin rays, etc.). Also, the discussion can be much clear if the proposed genera are used throughout the text (Aphanius mento group of species, inland and inland-related taxa, brackish water taxa, etc). If a new genus is erected, please be clear through the text and use it.

A preliminary hypothesis of phylogenetic relationships between genera can be proposed, using morphological characters and based on a valid molecular tree. Why not include it in this ms.?

Paraphanius has only 2 authors (line 353), it was proposed before?

Finally, please include full list of comparative material (again if the authors do not want to include it in the text, add supplementary material; lines 592-595).

6. PLOS authors have the option to publish the peer review history of their article (what does this mean?). If published, this will include your full peer review and any attached files.

Reviewer #1: No

Reviewer #2: No

---

## [Author Response · Author response to Decision Letter 0]

2 Mar 2020

Dear Prof. Roberta Cimmaruta,

Editor

Plos One

Many thanks for sending our manuscript PONE-D-19-33131 “DNA barcoding and species delimitation of the Old World tooth-carps, family Aphaniidae Hoedeman, 1949 (Teleostei: Cyprinodontiformes)” for revision. We also would like to thank the respected reviewers for their constructive comments on our manuscript. We found their comments and suggestions very constructive. So, we tried to implement all of them. We also provided some explanations for some of the comments. In all, we answered all the comments point by point, which are highlighted in yellow (please see below for the details of our answer to the comments). 

We also consider the following issues for submitting our revised manuscript:

• We provided a rebuttal letter as 'Response to Reviewers'. In this letter, we responded to each point raised by the academic editor and reviewer(s). This letter is uploaded as a separate file and labeled 'Response to Reviewers'.

• We provided a marked-up copy of our manuscript that highlights changes made to the original version. This file is uploaded as a separate file and labeled 'Revised Manuscript with Track Changes'.

• We provided an unmarked version of your revised paper without tracked changes. This file is uploaded as a separate file and labeled 'Manuscript'.

• The ms is edited by an English Native Person (B.W. Coad, Canadian Museum of Nature). 

I hope this ms is now in order and shall find a place in forthcoming issue of t Plos one journal.

Sincerey Yours

Prof. Hamid Reza Esmaeili

Details of the answer to the comments:

PONE-D-19-33131

DNA barcoding and species delimitation of the Old World tooth-carps, family Aphaniidae Hoedeman, 1949 (Teleostei: Cyprinodontiformes)

PLOS ONE

Dear Prof. Esmaeili,

Thank you for submitting your manuscript to PLOS ONE. After careful consideration, we feel that it has merit but does not fully meet PLOS ONE’s publication criteria as it currently stands. Therefore, we invite you to submit a revised version of the manuscript that addresses the points raised during the review process. 

Journal Requirements:

ANSWER: The manuscript meets the PLOS ONE's style requirements

2. In your Methods section, please provide additional location information of the sampling sites, including geographic coordinates for the data set if available.

ANSWER: The Data was included.

3. To comply with PLOS ONE submissions requirements, please provide methods of sacrifice in the Methods section of your manuscript.

ANSWER: The methods of sacrifice are added to the Methods section.

4. We noticed you still have some occurrence of overlapping text with the following previous publications, which needs to be addressed:

https://link.springer.com/article/10.1007%2Fs10641-009-9549-5

https://onlinelibrary.wiley.com/doi/full/10.1046/j.1420-9101.2003.00475.x

https duplicated text outside the methods section. Further consideration is dependent on these concerns being addressed.://www.biotaxa.org/Zootaxa/article/view/zootaxa.4338.3.10

In your revision ensure you cite all your sources (including your own works), and quote or rephrase any 

ANSWER: we changed the text, and also provide references to reduce the overlapping of the text with the previous publications. 

5. We note that Figure 1 in your submission contains map images which may be copyrighted. All PLOS content is published under the Creative Commons Attribution License (CC BY 4.0), which means that the manuscript, images, and Supporting Information files will be freely available online, and any third party is permitted to access, download, copy, distribute, and use these materials in any way, even commercially, with proper attribution. For these reasons, we cannot publish previously copyrighted maps or satellite images created using proprietary data, such as Google software (Google Maps, Street View, and Earth). For more information, see our copyright guidelines: http://journals.plos.org/plosone/s/licenses-and-copyright.

ANSWER: The map was originally designed in DIVA-GIS ver. 7.5, which is a free software. The information is now added to the manuscript. 

Reviewers' comments:

Reviewer's Responses to Questions

Comments to the Author

1. Is the manuscript technically sound, and do the data support the conclusions?

Reviewer #1: Yes

Reviewer #2: Partly

2. Has the statistical analysis been performed appropriately and rigorously?

Reviewer #1: Yes

Reviewer #2: No

3. Have the authors made all data underlying the findings in their manuscript fully available?

Reviewer #1: Yes

Reviewer #2: No

4. Is the manuscript presented in an intelligible fashion and written in standard English?

Reviewer #1: No

Reviewer #2: No

5. Review Comments to the Author

Reviewer #1: Dear Editor,

the manuscript PONE-D-19-33131 “DNA barcoding and species delimitation of the Old World tooth-carps, family Aphaniidae Hoedeman, 1949 (Teleostei: Cyprinodontiformes)” is an interesting paper dealing with the phylogenetic status of the peri-mediterranean genus Aphanius. The authors splitted up the species of this genus in the new genus Paraphanius, the genus Aphaniops (Hoedeman, 1951) and Aphanius Nardo, 1827, by applying four different molecular species delimitation methods to a consistent molecular dataset of cytochrome c oxidase I sequences. The methodology is appropriate and support the results and the main conclusion of the manuscript. The cited literature is quantitatively important and exhaustive and testifies to the authors' expertise. In my opinion this manuscript is of interest for the publication on Plos, but not in the present form because it needs major revision. 

First of all the text should be revised by a native English speaker. I found the text hard to read in many parts with sentences that could be written more clearly. However, I would like to stress that the discussion is much more fluent to read than the rest of the manuscript. The authors could lighten the text making it more effective for reading in some parts. ANSWER: The manuscript has been checked for its English. 

Line 27 relic should be relict

ANSWER: Many thanks for notifying this point. It is corrected now 

Line 39 “the family, including (i)…” “the family: (i) the first clade, positioned at the base of the phylogenetic tree, include A. mento and…..”

ANSWER: It is changed as requested by the respected reviewer. 

Line 42 “third cladecontains” “ third clade contains”

ANSWER: Many thanks for notifying this point. It is corrected now 

Lines 55-56 “In this concept, morphology-based methods have high efficiency.” This sentence could be deleted

ANSWER: It is deleted.

Line 57 “often lack the ability to discover the hidden species,” could be written “often does not allow to discover criptic species”

ANSWER: Many thanks. It is corrected as requested by the respected reviewer. 

Line 62 “DNA barcoding with…” “DNA barcoding based on…”

ANSWER: Many thanks. It is corrected as requested by the respected reviewer. 

Line 62 “ …cytochrome c oxidase I), COI(“ should be “…cytochrome c oxidase I (COI)”

ANSWER: It is corrected now. 

Line 64 “…with a low average distance within species of 0.39% [4].” This sentence could be changed “…only if the average intraspecific and interspecific genetic distance are significantly different (barcoding gap).

ANSWER: It is changed based on this comment of the respected reviewer. 

Lines 65-67 “DNA barcoding has been sufficiently used for the species identification because of the universal primers described by Ward et al. [4] and Ivanova et al. [5] that were very effective for the amplification of the COI sequences of most species” This sentece could be deleted.

ANSWER: Ok. it is done!

Line 68 Please delete however

ANSWER: Ok. it is done!

Line 68 “acuracyto” “accuracy to”

ANSWER: Many thanks for this notification. It is corrected now. 

Line 74 “along with” should be “along”

ANSWER: Ok. it is done!

Line 76 “..have largely been affected their..” should be “have largely affected their..”

ANSWER: Ok. it is done!

Line 76 “ distribution. .” delete the point

ANSWER: Ok. it is done!

Lines 77-81 this sentence must be written in a correct English!

ANSWER: Ok. it is done!

Lines 89-90 “Aphanius princeps is the taxon with the oldest fossil skeleton, and found in the deposits of Burdigalian age in Catalonia, NE Spain [8].” could be “Aphanius princeps, found in the deposits of Burdigalian age in Catalonia, NE Spain, is the taxon with the oldest fossil skeleton [8].”

ANSWER: Many thanks for this notification. It is corrected now. 

Lines 91-95 Please simplify this sentence “The genus Aphanius is the only native representatives, in the Old World, of the family Cyprinodontidae recently renamed Aphaniidae [9, 10-12], as firstly suggested by Sethi in 1960 [13]"

ANSWER: OK. It is done!

Line 111 Please include the reference to Lazara.

ANSWER: OK. It is done!

Line 113 “La Cepede” should be “Lacépède”

ANSWER: OK. It is done!

Line 117 “…however Aphanius dispar….” Should be “…while Aphanius dispar…”

ANSWER: OK. It is done!

Lines 119- 122 and 123-128 These two sentences are repetitive. The authors should keep one of them.

ANSWER: The second one is the key for the discrimination of Aphanius from Aphaniops.

Line 138-175 The information of this paragraph could be summarized on a table.

This was also our first decision, but with respect to this comment of the reviewer, since this paragraph has a large text data, it was a bit difficult to put all the data in a table. That was a reason that we decided to bring it to the main text. So, we would keep it as it is if the respected reviewer does not mind. 

Line 180 “..represents..” should be “..host..”

ANSWER: OK. It is done!

Line 186 “…up until recently..” could be deleted

ANSWER: OK. It is done!

Line 193”..delamination..” should be “delimitation”

ANSWER: OK. It is done!

Line 194 Delete “for”

ANSWER: OK. It is done!

Lines 195-198 “For this……of these fishes” Delete this ripetitive sentence

ANSWER: OK. It is done!

Line 288 Delete “was”

ANSWER: OK. It is done!

Line 301, 319 and 325“….delimited obviously..” Line 310 “..delimited nicely..” In don’t understand why the authors use these adverbs.

ANSWER: We revised this part for better understanding as the respected reviewer requested. 

Lines 332 “.. genetic distance in the COI barcode region the following groups..” should be “.. genetic distance in the COI barcode region for the following groups..”

ANSWER: OK. It is done!

Lines 412-413 Delete “..also lack the ability..”

ANSWER: OK. It is done!

Lines 416-417 “Therefore recent taxonomic and phylogenetic studies are based on morphological and molecular data”

ANSWER: OK. It is done!

Line 420 “..and compared the results..” should be “ ..and to compare the results..”

ANSWER: OK. It is done!

Line 436 “according to the temporal diversification…” “ according to their temporal diversification.”

ANSWER: OK. It is done!

Line 436 “..A. isfahanensis contains..” “..A. isfahanensis containing..”

ANSWER: OK. It is done!

Line 438 “..probably diverged..” “..which probably diverged..”

ANSWER: OK. It is done!

Line 445 “..subclade are distinct” : please add a reference

ANSWER:The references are now added

Line 461 “…length height value..” …lenght/height va

ANSWER: OK. It is done!

Reviewer #2: DNA barcoding and species delimitation of the Old World tooth-carps, family

Aphaniidae Hoedeman, 1949 (Teleostei: Cyprinodontiformes), Esmaeili et al.

This is an interesting manuscript that uses COI barcoding as a method for species delimitation within the Aphaniidae and proposes a new genus, using both published and new molecular data. The objectives of this ms. are 1) to use DNA barcoding as a tool for species delimitation within the family Aphaniidae; 2) to give a clearer picture of intra-family relationships; and 3) to clarify the generic names for the species groups within the family.

As presented, this manuscript needs to improve a few issues before can be accepted for publication.

First of all, only 14 species were identified (lines 212-213) and sequenced by the authors, while the others sequences belong to published information in part by the same group of authors. Unfortunately, not always the species identification of sequences deposited in Genbank is correct, and this might explain partially the disagreement between current taxonomy and the phylogenetic analysis. I suggest to clarify/explain the identification of species done by the authors and based in morphological diagnostic characters. If diagnostic characters are not external, then species identification might not be reliable (as specimens need to be cleared and stained for example). If some individuals were not identified, either by the authors or others, then these individuals should be clearly marked/identified. Another suggestion is to separate the analysis in two: only considering information gathered by the authors (i.e. information that the authors can trust), and a second analysis including published sequences.

ANSWER: The genus Aphanius is widely distributed. We sampled 14 species from various localities in Iran and presented their COI sequence for the first time (Fig. 1). This comment of the respected reviewer could, of course, be right. As the respected reviewer probably knows, other Aphanius species are distributed in geographic regions, which are politically difficult to do sampling. It is not only a problem for us but also for other researchers. Another important issue is that many of these sequences obtained from NCBI are the results of several previous studies that are published in good international journals. We also checked their analysis to be sure about the validity of the results. Therefore, this was a point that we think that the sequences from NCBI for this study are more likely correct and could be used for analysis. This outcome could also be supported by our study because we found that all the sequences that we extracted from the NCBI have been clustered in the correct place in the phylogenetic analysis. I hope this explanation would be ok to convince the respected reviewer. 

Both species identification (lines 212-213) and molecular methods (220-223) are too brief. If the authors do not want to include full methods within the text please add a supplement including details of morphological characters for species identification and methods that clarify molecular tasks. The main results of this ms. are based in these methods, so they should be included in this ms. and not citing third party or other papers from the authors. It is really important to explain the readers how the individuals were identified and how the sequences were obtained.

ANSWER: To consider this comment of the respected reviewer, we added some information into the text with regard to the molecular methods and morphological identification of the species. However, we would like to give this information to the respected reviewer that, the Iranian Aphanius species were identified based on the original descriptions, most of these species (60%) are described by H.R. Esmaili and A. Teimori: Esmaeili et al. (2014), Coad (1988), Hrbek (2006), Coad (2009), Teimori et al. (2011), Esmaeili et al. (2012), Teimori et al. (2012), Gholami et al. (2014), Teimori et al. (2014) and Coad (1980). Full details of morphological descriptions and diagnostics for these Aphanius species are already published in the scientific journals. This is the common methodology and is consistent with other major DNA barcoding studies already published in Plos ONE (e.g. Díaz et al., 2016; Bingpeng et al., 2018). This is also the common methodology followed by other top journals such as Scientific Reports (e.g. Guimarães-Costa et al., 2019; Katouzian et al., 2016).

The COI gene might not be a good marker for phylogenetic analysis, as it might be saturated. Please include an analysis showing that the analysis is not skewed due to saturation (e.g. using DAMBE [Xia & Xie 2001, J Hered 92(4):371-3] or other software). If this is the case, then the phylogenetic analysis cannot be presented as is, and the authors should refer to it as a phenetic analysis. In this regard, the outgroup used might not be adequate, please include other species/genera as there is no mention for the systematic account of the Aphaniidae and relatives. It should be noted that the new sequences are not deposited in Genbank yet, and thus are not accessible (line 234).

ANSWER: Thank you for giving this comment. We examined the substitution saturation with DAMBE ver. 7.2.7 using the Xia et al., (2013), and Xia and Lemey (2009) test of substitution saturation. The nucleotide substitution pattern showed that the sequences have not reached substitution saturation and are therefore well applicable for phylogenetic analyses. In the case of codon positions 1 and 2, we obtain Iss= 0.042, much smaller than Iss.c (= 0.789 assuming a symmetrical topology and 0.757 assuming an asymmetrical topology). In the case of codon position 3, we obtain Iss= 0.488, much smaller than Iss.c (= 0.776 assuming a symmetrical topology and 0.767 assuming an asymmetrical topology). Based on these results, the COI sequences obviously have experienced only little substitution saturation and consequently contain significant phylogenetic information. We accordingly, revised the text. 

Considering the fossil calibration (lines 245-247), the results should be cautionary interpreted. Have the authors calculated a mutation rate? It seems that it is much higher than “standard” rates of 1-3% per million year. If this is the case, then the ultrametric tree produced by BEAST might be biased and I suggest not to use it (or to change/delete the fossil date for calibration). Also, BEAST produce a credible interval for each node: are these broad? These re-analyses might change completely the molecular results and interpretation. 

ANSWER: bGMYC is conceptually similar to bPTP and uses a tree topology to infer species hypotheses, but unlike bPTP, it applies an ultrametric tree as an input file. To date, no mutation rate has been reported for COI in Aphanius. Therefore, to produce an ultrametric tree we followed Teimori et al. (2018) which is based on fossil calibration: Teimori, A., Esmaeili, H.R., Hamidan, N. and Reichenbacher, B., 2018. Systematics and historical biogeography of the Aphanius dispar species group (Teleostei: Aphaniidae) and description of a new species from Southern Iran. Journal of Zoological Systematics and Evolutionary Research, 56(4), pp.579-598. 

It should be noted that the K2P distances are used and not phylogenetic corrected distances (lines 346-351). These are phenetic but not phylogenetic results. Also, please see Srivathsan & Meier (2012). On the inappropriate use of Kimura-2-parameter (K2P) divergences in the DNA-barcoding literature. Cladistics 28, 190–194.

ANSWER: Srivathsan & Meier (2012) demonstrated that K2P is not an appropriate model for closely related sequences, but please consider that our three groups of sequences (i.e. Aphanius, Aphaniops and Paraphanius) highly diverge.

Figure 2 include two numbers at some nodes (posterior probabilities and bootstrap values) but not for all of them. Why not every single node has these 2 values? This mean that a node was not recovered with both methodologies? 

ANSWER: All these nodes were recovered with both methodologies but the posterior probabilities and bootstrap values lower than 0.5 and 50% are not shown at these nodes. Meanwhile, we noticed some missing values as the respected reviwer has mentioned correctly. We edited the figure once again, and the new tree is submitted now.

Please discuss the validity of the support at each node, especially if some nodes were not recovered with both approaches.

One of the strengths of this ms. is that the authors seek for an agreement between morphological characters and a molecular approach which is one of the main contributions of the ms. I suggest to include a table where the reader can easily understand if each species has a valid diagnosis (whether the characters are internal or external) and if according to the molecular methodology also would be a valid species. Sometimes the results are not clear enough and this addition might improve both the quality and clarity of the ms. 

ANSWER: Many thanks for providing this comment. The morphological characteristics of valid species are given in the recent published articles (e,g., Yoğurtçuoğlu et al., 2018; Freyhof et al., 2017; Freyhof et al. 2017; Pfleiderer et al., 2014; Teimori et al., 2014; Gholami et al., 2014; Coad, 1988; ) and even the keys have been provided. All of these references and seveal others have been cited in this ms. Hence we think that adding these information again here will just incease the length of ms.

Please also discuss (and add to the methodology) which species concept each of the 4 molecular methods stand for (see De Queiroz, Syst. Biol. 56, 879–886 [2007]). Under some species concepts (e.g. monophyly) one species might not be valid but can be diagnosable. This highlight the fact that some morphological characters are based in males and COI gene has maternal inheritance, please discuss this if that is occurring.

ANSWER: We added information into the text to consider this comment of the respected reviewer. 

Regarding morphological characters and specifically the systematics of the 3 genera proposed under Aphaniidae, it is not completely clear which characters are diagnostic, and which belong to the description of each genus (lines 362-363: characters overlap between genera, e.g. dorsal fin rays; and are not described for Aphanius; lines 543-551: number of total vertebrae, principle caudal-fin rays, etc.). ANSWER: Yes, you are right, but we summarized all the characters that seem to be different between the two genera and the key which shows diahnistic characteristics are provided.. Also for some we bring a range to show some difference for them. We also changed the text for a better understanding of the point. Also, the discussion can be much clear if the proposed genera are used throughout the text (Aphanius mento group of species, inland and inland-related taxa, brackish water taxa, etc). If a new genus is erected, please be clear through the text and use it. 

ANSWER: Many thanks for providing this comment. We changed the text to consider this comment from the respected reviewer. 

A preliminary hypothesis of phylogenetic relationships between genera can be proposed, using morphological characters and based on a valid molecular tree. Why not include it in this ms?

ANSWER: A preliminary hypothesis of phylogenetic relationships between genera is already proposed in the discussion (see section “Taxonomic remark on the genus Aphanius Nardo, 1827”).

Paraphanius has only 2 authors (line 353), it was proposed before?

ANSWERE: 

Paraphanius is a new genus proposed for the first time in this study, and the author names have been added now. It is done

Finally, please include full list of comparative material (again if the authors do not want to include it in the text, add supplementary material; lines 592-595).

ANSWER:

As the data set are too large hence, and all done by authors in the previous published articles, the comparative materials are referenced in the manuscript:

See lists of materials examined by Esmaeili et al. ([50]: farsicus), Esmaeili et al. ([19]: A. darabensis and A. kavirensis); Teimori et al. ([47]: A. arakensis), Gholami et al. ([48]: A. shirini); Gholami et al. ([20, 80]: A. farsicus and A. sophiae), Teimori et al. ([51]: A. furcatus) and Teimori et al. ([15]: A. dispar group). 

The ms is now edited by an English native Persion (Dr. Brian W. Coad/ ichthyologist ) from the Canadian Museum of nature.

---

## [Decision Letter · Decision Letter 1]

26 Mar 2020

PONE-D-19-33131R1

DNA barcoding and species delimitation of the Old World tooth-carps, family Aphaniidae Hoedeman, 1949 (Teleostei: Cyprinodontiformes)

PLOS ONE

Dear Prof. Esmaeili,

Thank you for submitting the improved version of your  manuscript. Many suggested changes have been implemented,  but some more work is needed fully meet PLOS ONE’s publication criteria. Therefore, we invite you to address all the points raised by the reviewer before re-submitting.

In particular, the parts concerning morphological analyses and fossil calibration must be changed according to the suggestions provided.  

We would appreciate receiving your revised manuscript by May 10 2020 11:59PM. To enhance the reproducibility of your results, we recommend that if applicable you deposit your laboratory protocols in protocols.io, where a protocol can be assigned its own identifier (DOI) such that it can be cited independently in the future. For instructions see: http://journals.plos.org/plosone/s/submission-guidelines#loc-laboratory-protocols

We look forward to receiving your revised manuscript.

Kind regards,

Roberta Cimmaruta, PhD

Academic Editor

PLOS ONE

Reviewers' comments:

Reviewer's Responses to Questions

**Comments to the Author**

1. If the authors have adequately addressed your comments raised in a previous round of review and you feel that this manuscript is now acceptable for publication, you may indicate that here to bypass the “Comments to the Author” section, enter your conflict of interest statement in the “Confidential to Editor” section, and submit your "Accept" recommendation.

Reviewer #2: (No Response)

2. Is the manuscript technically sound, and do the data support the conclusions?

Reviewer #2: Yes

3. Has the statistical analysis been performed appropriately and rigorously? 

Reviewer #2: Yes

4. Have the authors made all data underlying the findings in their manuscript fully available?

Reviewer #2: Yes

5. Is the manuscript presented in an intelligible fashion and written in standard English?

Reviewer #2: Yes

6. Review Comments to the Author

Reviewer #2: DNA barcoding and species delimitation of the Old World tooth-carps, family Aphaniidae Hoedeman, 1949 (Teleostei: Cyprinodontiformes), Esmaeili et al.

This is the second time I review this manuscript. I thank the authors for the modifications and changes to the ms., but I still have some recommendations that will improve the quality of the manuscript. Please consider that some of the changes were difficult to understand, either because the response was not detailed (i.e. not stating the lines modified) or simply because they were not considered.

I will add a few suggestions following the original order of comments, especially those comments that were not considered by the authors (or were not justified).

Line 203. Change “Sampling and morphological identification” to “Sampling and external morphological identification”

If meristic counts of fin rays were not done in cleared and stained specimens, please change: “based on the external morphology, including coloration and meristic counts” to “based on the external morphology and including coloration.” (lines 207-208). Then add to the end of that paragraph “Meristic counts were done externally (not using cleared and stained specimens).” (line 210). To the knowledge of this reviewer, fin rays count should always be done in cleared and stained specimens and not externally.

Line 275. Add “belonging to morphologically identified species based on external characters”

As I mentioned in my previous review, every manuscript should be self-contained. Please include temperature cycle and which Taq/reagents/mastermix was used for PCR (lines 223-225).

Lines 239-241. Please move the phrase “The nucleotide substitution pattern showed that the sequences have not reached substitution saturation and are therefore well applicable for phylogenetic analyses.” to results section.

In my previous review I mentioned that considering the fossil calibration used (maximum age 34 Ma) (lines 307-309), the results should be cautionary interpreted. As no mutation rate has been reported for COI in Aphanius, a brief paragraph discussing it should be included (please consider credible intervals for genera and some important species-group nodes at least).

Finally, I recall that every manuscript should be self-contained. Please include full list of comparative material as supplementary material, so that the length of the manuscript is not increased.

7. PLOS authors have the option to publish the peer review history of their article (what does this mean?). If published, this will include your full peer review and any attached files.

Reviewer #2: No

---

## [Author Response · Author response to Decision Letter 1]

28 Mar 2020

Dear Prof. Roberta Cimmaruta

Academic Editor 

Plos One

Many thanks for sending our manuscript PONE-D-19-33131R1

“DNA barcoding and species delimitation of the Old World tooth-carps, family Aphaniidae Hoedeman, 1949 (Teleostei: Cyprinodontiformes)” for the second round of revision. We also would like to thank the respected reviewer for his/her constructive comments on our manuscript. We found the comments and suggestions very constructive. So, we implemented all of them. 

We also considerd the following issues for submitting our revised manuscript:

• A rebuttal letter that responds to each point raised by the academic editor and reviewer(s). This letter was uploaded as separate file and labeled 'Response to Reviewers'.

• A marked-up copy of the second revised manuscript that highlights changes made to the original version. This file was uploaded as separate file and labeled 'Revised Manuscript with Track Changes'.

• An unmarked version of of the second revised paper without tracked changes. This file was uploaded as separate file and labeled 'Manuscript'.

• The figure files were uploaded to the Preflight Analysis and Conversion Engine (PACE) digital diagnostic tool, https://pacev2.apexcovantage.com/ to be ensured that figures met PLOS requirements.

We also made very few minor grammatical corrections. These are clearely indicated using Track Change System.

I hope this revised manuscript is now in order and shall find a place in the forthcoming issue of the Plos One journal.

One again, I ihank you for handling the ms.

Sincerely Yours

Details of the answer to the comments:

PONE-D-19-33131R1

“DNA barcoding and species delimitation of the Old World tooth-carps, family Aphaniidae Hoedeman, 1949 (Teleostei: Cyprinodontiformes) ”

The main points suggested by respected reviewer:

#Line 203. Change “Sampling and morphological identification” to “Sampling and external morphological identification”

ANSWER: Corrected: It was changed to: Sampling and external morphological identification

#If meristic counts of fin rays were not done in cleared and stained specimens, please change: “based on the external morphology, including coloration and meristic counts” to “based on the external morphology and including coloration.” (lines 207-208). 

ANSWER: Corrected: It was changed to “based on the external morphology and including coloration”

#Then add to the end of that paragraph “Meristic counts were done externally (not using cleared and stained specimens).” (line 210). 

ANSWER: Corrected: the following sentence was added to the end of paragraph. 

“Meristic counts were done externally (not using cleared and stained specimens)”

To the knowledge of this reviewer, fin rays count should always be done in cleared and stained specimens and not externally.

#Line 275. Add “belonging to morphologically identified species based on external characters”

ANSWER: It was in line 220: 

Corrected: belonging to morphologically identified species based on external characters”

#As I mentioned in my previous review, every manuscript should be self-contained. Please include temperature cycle and which Taq/reagents/mastermix was used for PCR (lines 223-225).

ANSWER: The following sentence was added: 

The amplification process was performed using Master Mix in a total volume of 25 μl containing 12.5 μl of a Ready 2X PCR Master Mix (Genetbio, Cat. no. G-2000), 0.5 μl of each primer (10 pmol/μl), 5 μl of the DNA template and 6.5 μl dd water. The amplification was performed on a Bioer XP Thermal Cycler (Bioer Technology Co. Ltd., Hangzhou, China), programmed as following: an initial denaturation at 94 °C for 3 min, 35 cycles with denaturation at 94 °C for 50 s, annealing at 52 °C for 1min, and a final extension phase at 72 °C for 5 min.

#Lines 239-241. Please move the phrase “The nucleotide substitution pattern showed that the sequences have not reached substitution saturation and are therefore well applicable for phylogenetic analyses.” to results section.

ANSWER: Done: the phrase was moved to the results section under the heading of Molecular species delimitation (lines 299-300).

#In my previous review I mentioned that considering the fossil calibration used (maximum age 34 Ma) (lines 307-309), the results should be cautionary interpreted. As no mutation rate has been reported for COI in Aphanius, a brief paragraph discussing it should be included (please consider credible intervals for genera and some important species-group nodes at least).

ANSWER: It was done: the following sentence was added at the end of line 343.

The bGMYC method is similar to bPTP, but it uses an ultrametric tree to delimit species. As no mutation rate has been reported for COI in genus Aphanius, the maximum age for Aphanius was set at 34 Ma based on the oldest known fossil of the Old World killifishes to produce an ultrametric tree. For this reason, and since credible intervals for some species-group nodes were broad (not shown), the bGMYC results should be cautionary interpreted.

#Finally, I recall that every manuscript should be self-contained. Please include full list of comparative material as supplementary material, so that the length of the manuscript is not increased.

ANSWER: Done. Full list of comparative materials is listed in the supplementary material:

As: 

S1 Appendix. List of comparative material examined.

Additional change:

We registered and received LSID for the genus Paraphanius from ZooBank and thus the following statement was added:

Nomenclatural acts 

The electronic edition of this article conforms to the requirements of the amended International Code of Zoological Nomenclature, and hence the new names contained herein are available under that Code from the electronic edition of this article. This published work and the nomenclatural acts it contains have been registered in ZooBank, the online registration system for the ICZN. The ZooBank LSIDs (Life Science Identifiers) can be resolved and the associated information viewed through any standard web browser by appending the LSID to the prefix “http://zoobank.org/”. The LSID for this publication is: urn:lsid:zoobank.org:pub:A8F5EA21-50CE-4FBE-969C-EA6B61FFBABE

The mina letter sent by respected Editor:

Dear Prof. Esmaeili,

Thank you for submitting the improved version of your manuscript. Many suggested changes have been implemented, but some more work is needed fully meet PLOS ONE’s publication criteria. Therefore, we invite you to address all the points raised by the reviewer before re-submitting.

In particular, the parts concerning morphological analyses and fossil calibration must be changed according to the suggestions provided. 

We would appreciate receiving your revised manuscript by May 10 2020 11:59PM. To enhance the reproducibility of your results, we recommend that if applicable you deposit your laboratory protocols in protocols.io, where a protocol can be assigned its own identifier (DOI) such that it can be cited independently in the future. For instructions see: http://journals.plos.org/plosone/s/submission-guidelines#loc-laboratory-protocols

• A rebuttal letter that responds to each point raised by the academic editor and reviewer(s). This letter should be uploaded as separate file and labeled 'Response to Reviewers'.

• A marked-up copy of your manuscript that highlights changes made to the original version. This file should be uploaded as separate file and labeled 'Revised Manuscript with Track Changes'.

• An unmarked version of your revised paper without tracked changes. This file should be uploaded as separate file and labeled 'Manuscript'.

 We look forward to receiving your revised manuscript.

Kind regards,

Roberta Cimmaruta, PhD

Academic Editor

PLOS ONE

Reviewers' comments:

Reviewer's Responses to Questions

Comments to the Author

1. If the authors have adequately addressed your comments raised in a previous round of review and you feel that this manuscript is now acceptable for publication, you may indicate that here to bypass the “Comments to the Author” section, enter your conflict of interest statement in the “Confidential to Editor” section, and submit your "Accept" recommendation.

Reviewer #2: (No Response)

2. Is the manuscript technically sound, and do the data support the conclusions?

Reviewer #2: Yes

3. Has the statistical analysis been performed appropriately and rigorously?

Reviewer #2: Yes

4. Have the authors made all data underlying the findings in their manuscript fully available?

Reviewer #2: Yes

5. Is the manuscript presented in an intelligible fashion and written in standard English?

Reviewer #2: Yes

6. Review Comments to the Author

Reviewer #2: DNA barcoding and species delimitation of the Old World tooth-carps, family Aphaniidae Hoedeman, 1949 (Teleostei: Cyprinodontiformes), Esmaeili et al.

This is the second time I review this manuscript. I thank the authors for the modifications and changes to the ms., but I still have some recommendations that will improve the quality of the manuscript. Please consider that some of the changes were difficult to understand, either because the response was not detailed (i.e. not stating the lines modified) or simply because they were not considered.

I will add a few suggestions following the original order of comments, especially those comments that were not considered by the authors (or were not justified).

#Line 203. Change “Sampling and morphological identification” to “Sampling and external morphological identification”

Corrected: It was changed to: Sampling and external morphological identification

#If meristic counts of fin rays were not done in cleared and stained specimens, please change: “based on the external morphology, including coloration and meristic counts” to “based on the external morphology and including coloration.” (lines 207-208). 

Corrected: It was changed to “based on the external morphology and including coloration”

#Then add to the end of that paragraph “Meristic counts were done externally (not using cleared and stained specimens).” (line 210). 

Corrected: the following sentence was added to the end of paragraph. 

“Meristic counts were done externally (not using cleared and stained specimens)”

To the knowledge of this reviewer, fin rays count should always be done in cleared and stained specimens and not externally.

#Line 275. Add “belonging to morphologically identified species based on external characters”

It was in line 220: 

Corrected: belonging to morphologically identified species based on external characters”

#As I mentioned in my previous review, every manuscript should be self-contained. Please include temperature cycle and which Taq/reagents/mastermix was used for PCR (lines 223-225).

The following sentence was added: 

The amplification process was performed using Master Mix in a total volume of 25 μl containing 12.5 μl of a Ready 2X PCR Master Mix (Genetbio, Cat. no. G-2000), 0.5 μl of each primer (10 pmol/μl), 5 μl of the DNA template and 6.5 μl dd water. The amplification was performed on a Bioer XP Thermal Cycler (Bioer Technology Co. Ltd., Hangzhou, China), programmed as following: an initial denaturation at 94 °C for 3 min, 35 cycles with denaturation at 94 °C for 50 s, annealing at 52 °C for 1min, and a final extension phase at 72 °C for 5 min.

#Lines 239-241. Please move the phrase “The nucleotide substitution pattern showed that the sequences have not reached substitution saturation and are therefore well applicable for phylogenetic analyses.” to results section.

Done: the phrase was moved to the results section under the heading of Molecular species delimitation (lines 299-300).

#In my previous review I mentioned that considering the fossil calibration used (maximum age 34 Ma) (lines 307-309), the results should be cautionary interpreted. As no mutation rate has been reported for COI in Aphanius, a brief paragraph discussing it should be included (please consider credible intervals for genera and some important species-group nodes at least).

It was done: the following sentence was added at the end of line 343.

The bGMYC method is similar to bPTP, but it uses an ultrametric tree to delimit species. As no mutation rate has been reported for COI in genus Aphanius, the maximum age for Aphanius was set at 34 Ma based on the oldest known fossil of the Old World killifishes to produce an ultrametric tree. For this reason, and since credible intervals for some species-group nodes were broad (not shown), the bGMYC results should be cautionary interpreted.

#Finally, I recall that every manuscript should be self-contained. Please include full list of comparative material as supplementary material, so that the length of the manuscript is not increased.

 Done. Full list of comparative materials is listed in the supplementary material:

As: 

S1 Appendix. List of comparative material examined.

7. PLOS authors have the option to publish the peer review history of their article (what does this mean?). If published, this will include your full peer review and any attached files.

Do you want your identity to be public for this peer review? For information about this choice, including consent withdrawal, please see our Privacy Policy.

Reviewer #2: No

It was done.

We got LSID for Paraphanius and thus the following statement was added:

Nomenclatural acts 

The electronic edition of this article conforms to the requirements of the amended International Code of Zoological Nomenclature, and hence the new names contained herein are available under that Code from the electronic edition of this article. This published work and the nomenclatural acts it contains have been registered in ZooBank, the online registration system for the ICZN. The ZooBank LSIDs (Life Science Identifiers) can be resolved and the associated information viewed through any standard web browser by appending the LSID to the prefix “http://zoobank.org/”. The LSID for this publication is: urn:lsid:zoobank.org:pub:A8F5EA21-50CE-4FBE-969C-EA6B61FFBABE

---

## [Editor Report · Decision Letter 2]

31 Mar 2020

DNA barcoding and species delimitation of the Old World tooth-carps, family Aphaniidae Hoedeman, 1949 (Teleostei: Cyprinodontiformes)

PONE-D-19-33131R2

Dear Dr. Esmaeili,

We are pleased to inform you that your manuscript has been judged scientifically suitable for publication and will be formally accepted for publication once it complies with all outstanding technical requirements.

With kind regards,

Roberta Cimmaruta, PhD

Academic Editor

PLOS ONE
---

## [Editor Report · Acceptance letter]

3 Apr 2020

PONE-D-19-33131R2 

DNA barcoding and species delimitation of the Old World tooth-carps, family Aphaniidae Hoedeman, 1949 (Teleostei: Cyprinodontiformes) 

Dear Dr. Esmaeili:

I am pleased to inform you that your manuscript has been deemed suitable for publication in PLOS ONE. Congratulations! Your manuscript is now with our production department. 

With kind regards,

on behalf of

Professor Roberta Cimmaruta 

Academic Editor

PLOS ONE